# M3: 3D-Spatial Multimodal Memory

**Xueyan Zou**[1◇], **Yuchen Song**[1◇], **Ri-Zhao Qiu**[1◇], **Xuanbin Peng**[1◇]
**Jianglong Ye**[1], **Sifei Liu**[2], **Xiaolong Wang**[1,2]
◇Core Contribution
[1]UC San Diego [2]NVIDIA
https://m3-spatial-memory.github.io

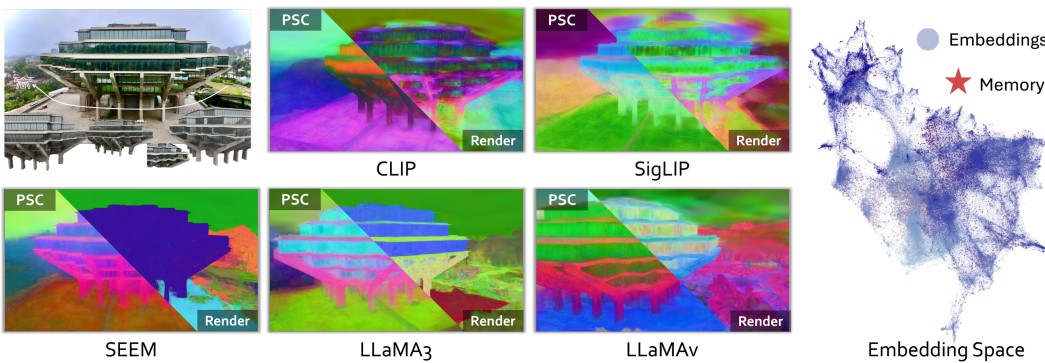

Figure 1: Our proposed MultiModal Memory integrates Gaussian splatting with foundation models to efficiently store multimodal memory in a Gaussian structure. The feature maps rendered by our approach exhibit high fidelity, preserving the strong expressive capabilities of the foundation models.

## ABSTRACT

We present 3D Spatial MultiModal Memory (**M3**), a multimodal memory system designed to retain information about medium-sized static scenes through video sources for visual perception. By integrating 3D Gaussian Splatting techniques with foundation models, **M3** builds a multimodal memory capable of rendering feature representations across granularities, encompassing a wide range of knowledge. In our exploration, we identify two key challenges in previous works on feature splatting: (1) computational constraints in storing high-dimensional features for each Gaussian primitive, and (2) misalignment or information loss between distilled features and foundation model features. To address these challenges, we propose **M3** with key components of principal scene components and Gaussian memory attention, enabling efficient training and inference. To validate **M3**, we conduct comprehensive quantitative evaluations of feature similarity and downstream tasks, as well as qualitative visualizations to highlight the pixel trace of Gaussian memory attention. Our approach encompasses a diverse range of foundation models, including vision-language models (VLMs), perception models, and large multimodal and language models (LMMs/LLMs). Furthermore, to demonstrate real-world applicability, we deploy **M3**'s feature field in indoor scenes on a quadruped robot. Notably, we claim that **M3** is the first work to address the core compression challenges in 3D feature distillation.

## 1 INTRODUCTION

Human perception encompasses the world across spatial dimensions. When encountering static elements in the visual environment, individuals tend to organize and store knowledge progressively, starting from a coarse overview and refining it into finer details. For instance, we can recall our daily surroundings with varying levels of detail, ranging from a high-level layout to specific part-level features. However, for larger-scale environments, our understanding tends to remain more coarse and generalized. Previous works, such as NeRF [23] and 3DGS [16], have demonstrated the

ability to store scene-level information at the pixel level for intermediate-scale scenes. However, these models lack the capability to retain the semantic understanding of the scene like humans.

In this study, we aim to develop a spatial memory system for static scenes, capable of processing static scene video clips spanning spatial horizons. The primary objective is to store all human-processable information in a format that is precise, efficient, and amenable to future interactive queries. Our approach leverages 3D Gaussian splatting techniques and incorporates features extracted from foundation models to construct scenes imbued with semantic knowledge. The selection of Gaussian splatting as our structural format was motivated by two key considerations: First, the need to address video redundancy through efficient compression, and second, the requirement for multi-granular information representation. Gaussian splatting inherently provides a framework for representing the smallest units of information as Gaussian primitives as well as naturally eliminating the spatial redundancy, aligning well with the motivations.

Previous feature splatting works such as F-3DGS [51] and F-Splat [26] directly distill 2D feature maps obtained from foundation models into 3D Gaussians via differentiable rendering. We observe two key issues: First, due to the computational limitations, the feature vector dimensions in Gaussian primitives are significantly reduced compared to the original 2D feature maps (typically 16-64 versus 1024), potentially causing an information bottleneck. Second, while the original feature maps may not be inherently 3D-consistent, enforcing 3D consistency in the Gaussians can cause misalignment between the original and distilled features. Consequently, the distilled feature may not accurately capture the knowledge embedded in the foundation model.

To address these issues, we present MultiModal Memory (**M3**), a better integration of Gaussian splatting and multimodal foundation models that efficiently store expressive multimodal memory in a Gaussian structure, facilitating spatial queries. Specifically, we propose to store the original high-dimensional 2D feature maps in a memory bank called principal scene components and use the low-dimensional principal queries from 3D Gaussians as indices. Instead of directly distilling the 2D features into 3D embeddings, we apply Gaussian memory attention between the principal scene components and principal queries to render the foundation model embeddings in a 3D scene.

In this way, we combine the best of both foundation models and Gaussian splatting: preserving the high expressive ability of the original foundation model feature maps while maintaining a 3D-consistent, low-dimensional Gaussian structure of the scene. Furthermore, we also design a heuristic algorithm to minimize redundancy in the memory bank by reducing the raw features from the video stream. These reduced features are referred to as Principal Scene Components. Example feature maps rendered by **M3** are visualized in Fig. 1.

To evaluate **M3**, we employ a diverse set of foundation models, including vision-language models, LMM/LLMs, and perception models. We adopt both low-level metrics (e.g. PSNR) to assess the model's feature memorization capability and high-level metrics (e.g. mIoU, IR, TR) to assess its performance on downstream perception tasks. Extensive experiments demonstrate that **M3** outperforms previous works in both memorization and downstream tasks while maintaining low computational costs. Lastly, we deploy **M3** on a quadruped robot platform for grasping, showcasing its potential for real-world generalization from single-scene, multi-scene, and long-horizon tasks.

## 2 RELATED WORK

**Foundation Models.** The field of multimodal learning has seen remarkable progress, leading to the development of diverse foundation models. In the vision-language domain, models such as CLIP [28], Florence [40; 45], and the recent SigLIP [47] employ ViT-style [5] transformer architectures to align visual and linguistic representations. For vision-specific tasks, SAM [17; 29] excels in part-level clustering, while DINO [4; 24] advances self-supervised representation learning. In document understanding, LayoutLM [42; 41; 11] combines OCR and text classification for comprehensive document analysis. The language domain has seen significant advancements in reasoning capabilities, exemplified by the LLaMA [36; 37; 6] and Mistral [14; 15] series. While these works have pushed language reasoning to new heights, recent studies like [31; 35; 7] explore mixture-of-experts approaches to enhance visual representation learning in foundation models. These developments, along with advanced models such as ChatGPT [1] and Claude [2], form the backbone of modern Multimodal Large Language Models (MLLMs), paving the way for more sophisticated AI systems.

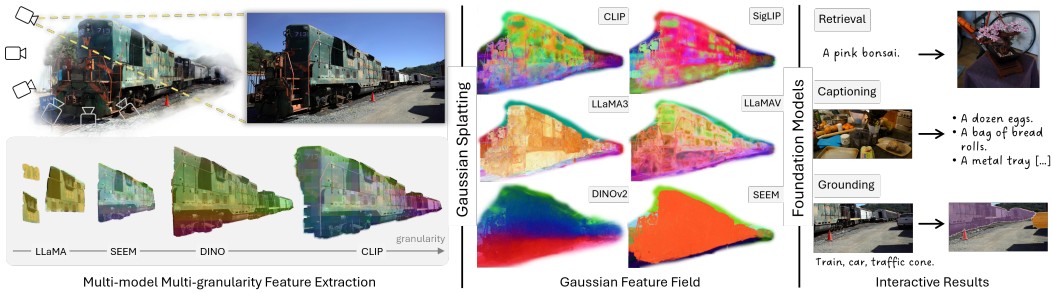

Figure 2: A scene (**V**) is composed of both structure (**S**) and knowledge (**I**). To model these, we leverage multiple foundation models to extract multi-granularity scene knowledge, and employ 3D Gaussian splatting to represent the spatial structure. By combining these techniques, we construct a spatial multimodal memory (**M3**), which enables downstream applications such as retrieval, captioning and grounding.

**3D Gaussians and Feature Field.** NeRF [23] revolutionized 3D scene representation, but its implicit nature caused slow rendering and training. 3D Gaussian Splatting [16] (3DGS) emerged as a faster, more explicit alternative, enabling rapid training and real-time rendering. Since then, 3DGS has been enhanced: H3DGS [44] improved large-scale rendering, Mip-Splatting tackled anti-aliasing for high detail, and WildGaussians [19] addressed occlusion and appearance changes. Grendel-GS [50] enabled multi-GPU training for efficiency with larger datasets. Researchers began incorporating 3D feature fields into neural rendering pipelines, moving from NeRF-based models like F3RM [32] to 3DGS-based ones. Feature 3DGS [51] added feature representations to 3DGS, leading to advancements like Feature Splatting [26; 13] for language-driven scene synthesis and LEGaussians [33] for open-vocabulary scene understanding. LiveScene [27] introduced interactive radiance fields, while recent work [46] focuses on improving 2D features with 3D-aware fine-tuning for better 2D-3D integration.

**Scene Graph and Video Memory.** Long-horizon scene understanding encompasses both spatial and temporal dimensions. For spatial modeling, scene graphs have been prominent: ConceptFusion [12] introduced open-set multimodal 3D mapping, ConceptGraphs [8] extended this to open-vocabulary 3D scene graphs, and Hierarchical Open-Vocabulary 3D Scene Graphs [39] applied these concepts to language-grounded navigation. Beyond Bare Queries [21] and Open Scene Graphs [22] further demonstrated their utility in object retrieval and navigation. However, these approaches often rely on heuristic edge/node construction and lack direct LMM integration via embeddings. For temporal aspects, previous works have focused on using memory bank embeddings to store information across frames. For instance, MA-LMM [9], MovieChat [34], and Hierarchical Memory [38] introduced various memory augmentation techniques for video understanding. Flash-VStream [48] and Streaming Long Video Understanding [25] concentrated on real-time processing of long video streams. While these temporal methods integrate better with LMMs, they face challenges such as image over-compression (representing an entire frame with a single embedding), frame redundancy (adjacent frames containing overlapping spatial information), and lack of explicit spatial information. Our 3D Gaussian approach bridges this gap, combining spatial precision with temporal flexibility and LMM compatibility.

## 3 METHOD

### 3.1 3D-SPATIAL MULTIMODAL MEMORY (M3) PIPELINE.

A real-world visual perception scene (**V**) consists of both structure (**S**) and knowledge (**I**). The structure of Visual Granularity ($\mathcal{VG}$) can range from the fine details such as leaf shapes, to large-scale elements, such as city layouts. Concurrently, the Knowledge Space ($\mathcal{KS}$) spans scales from specific information, such as leaf species (e.g. a red maple leaf) to a comprehensive interpretation (e.g. The space needle in Seattle...) of a view ($\mathbf{V}_*$). Gaussian splatting serves as a framework for constructing scene structure with finest granularity, represented as gaussian primitives, while foundation models provide vast world knowledge spanning various scales for scene knowledge. The organic integration of Gaussian splatting and Foundation Models infuses scene structure with multi-

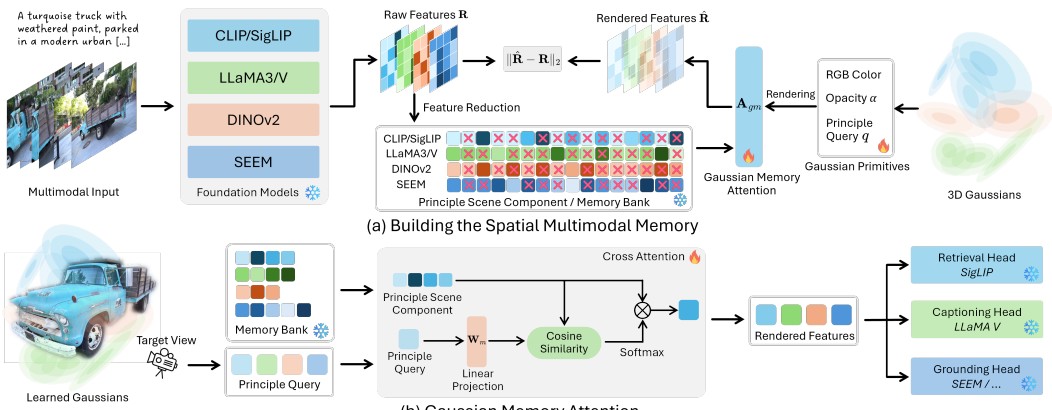

Figure 3: Given a video sequence, we utilize foundation models ($\mathbf{F}$) to extract raw features ($\mathbf{R}$). These features are reduced using Algorithm 1, producing principal scene components ($\mathbf{PSC}$), which are stored in a memory bank. We introduce optimizable attribute queries ($q$) to Gaussian primitives, and apply a Gaussian Memory Attention ($\mathbf{A}_{gm}$) mechanism to produce the final rendered features ($\hat{\mathbf{R}}$), which can be linked back to various heads of the foundation models.

granularity knowledge, enabling the construction of a full-stack Multimodal Memory of the scene with precise spatial information. To maintain efficiency while preserving the global representation of foundation model features, we compress the extracted features from foundation models into principal scene components ($\mathbf{PSC}$) for each scene and learn to probe the scene via Gaussian Splatting parameters, denoted as principle query ($\mathbf{Q}_p$). Ultimately, leveraging the rendering capabilities of Gaussian Splatting, we can dynamically populate the GS structure with multi-granularity information spanning the entire view of the scene. Our pipeline is illustrated in Fig. 2.

## 3.2 M3 Preliminaries.

**Visual Granularity ($\mathcal{VG}$).** Visual granularity ($\mathcal{VG}$) typically represents the clustering pixel scope of an image, a concept introduced in Semantic-SAM [20]. Given a view $\mathbf{V}_* \in \mathbb{R}^{h \times w, 3}$ ($h, w$ denote the pixel dimensions) in the scene $\mathbf{V}$, it is composed of multi-granularity segments ranging from individual pixels to the full view (as illustrated in the left part of Fig. 2), represented by $\mathbf{V}_* = \{V_*^1, V_*^2, ..., V_*^m\}$, where $V_*^i \in \mathbb{R}^{p,3}$ is the $i^{\text{th}}$ granularity of the view $\mathbf{V}_*$, $p$ is the number of pixels, and $m$ denotes the total number of granularities. This multi-granularity approach is introduced because humans naturally possess multi-granularity recognition of the world for various utilities.

**Knowledge Space ($\mathcal{KS}$).** Different foundation models ($\mathbf{F}$) focus on various aspects of knowledge. For instance, CLIP [28] and SigLIP [47] concentrate on image-level perception, while Semantic-SAM [20] emphasizes part-level visual grouping. In contrast, LLaMA3/v [6] incorporates both local and global attention mechanisms. The features generated by these models occupy different knowledge spaces $\mathbf{F}(\mathbf{V}_*) \in \{\mathcal{KS}^1, \mathcal{KS}^2, ..., \mathcal{KS}^c\}$ where $c$ is the total number of knowledge spaces here, emphasizing diverse aspects such as visual alignment ($\mathcal{KS}^1$), semantics ($\mathcal{KS}^2$), reasoning ($\mathcal{KS}^3$), and etc.

**Principle Scene Components (PSC) and Principle Query ($\mathbf{Q}_p$).** We extract foundation model features for each view, denoted as $\mathbf{F}_*(\mathbf{V}) = \{\mathbf{E}_1^*, \mathbf{E}_2^*, ...\mathbf{E}_n^*\}$ for each foundation model ($\mathbf{F}_*$) and scene ($\mathbf{V}$), where $n$ is the number of views. These foundation model features are represented as $\mathbf{E}_i^* \in \mathbb{R}^{[h \times w, d]}$, where $h, w$ denote the feature pixel dimensions. However, different views often contain redundant and similar features. We define the key features that construct the scene as Principle Scene Components ($\mathbf{PSC}$), drawing inspiration from the terminology of Principal Component Analysis. The attribute within Gaussian representation responsible for indexing $\mathbf{PSC}$ is denoted as Principle Query ($\mathbf{Q}_p$), which is learnable parameters in each Gaussian primitive.

## 3.3 Spatial Multimodal Memory

**Build Scene Structure via 3D Gaussians.** We formally define the input of **M3** as a video sequence with frames, where each frame corresponds to a view $\mathbf{V}_*$. 3D Gaussian splatting [16] is employed

---

**Algorithm 1** Raw Feature ($\mathbf{R}$) Similarity Reduction Algorithm

---

**Input** : $\mathbf{R} \in \mathbb{R}^{[n \times h \times w, d]}$ (Raw Features), $\theta \in (0, 1]$ (threshold), $c \in \mathbb{N}$ (chunk size)
**Output**: $\mathbf{PSC} \subseteq \mathbf{R}$ (Principle Scene Components)

1  SimilarityReduction($\mathbf{R}, \theta, c$) $n \leftarrow |\mathbf{R}|$ # Number of raw features
2  $\hat{\mathbf{R}} \leftarrow \{\frac{e_i}{\|e_i\|_2} : e_i \in \mathbf{R}\}$ # Normalize raw features
3  $I \leftarrow \emptyset$ # Set of filtered indices
4  $U \leftarrow \{0\}^n$ # Usage mask, initially all false
5  **for** $k \leftarrow 0$ **to** $\lfloor \frac{n}{c} \rfloor - 1$:
6      $C_k \leftarrow \{\hat{e}_i : i \in [kc, (k+1)c) \cap \mathbb{N}\}$ # Current chunk
7      $S_k \leftarrow C_k \cdot \hat{\mathbf{R}}^\top$ # Similarity matrix for chunk
8      **for** $j \leftarrow 0$ **to** $|C_k| - 1$:
9          **if** $U_{kc+j} = 0$:
10             $J \leftarrow \{i : S_{k,j,i} \geq \theta\}$ # Similar indices
11             **if** $\forall i \in J : U_i = 0$:
12                 $I \leftarrow I \cup \{kc + j\}$ # Select Principle Components
13                 $\forall i \in J : U_i \leftarrow 1$

14 $\mathbf{PSC} \leftarrow \{e_i : i \in I\}$ **return** $\mathbf{PSC}$

---

to fit the scene, with each view rendered by the Gaussian rasterizer. For each Gaussian primitive, the optimizable attributes include the centroid ($x \in \mathbb{R}^3$), rotation quaternion ($r \in \mathbb{R}^4$) storing the rotation and scaling matrix, opacity value ($\alpha \in \mathbb{R}^3$), and spherical harmonics ($sh \in \mathbb{R}^3$). To model the Principle Scene Components ($\mathbf{PSC}$), we introduce an additional optimizable attribute: principle queries ($q \in \mathbb{R}^l$) with flexible dimensionality to accommodate various foundation models. Each foundation model utilizes $s$ degrees from the $\mathbf{Q}_p \in \mathbb{R}^l$. These degrees are rendered alongside Gaussian parameters to produce view-based principle queries $\mathbf{Q}_p^{\mathbf{V}_*}$ with shape $[H, W, l]$. Following [51], the colors and principle queries are rendered as:

$$C = \sum_{i \in N} c_i \alpha_i T_i, \quad \mathbf{Q}_p = \sum_{i \in N} q_i \alpha_i T_i, \quad \text{where } T_i = \prod_{j=1}^{i-1}(1 - \alpha_j) \tag{1}$$

Here, $N$ represents the set of sorted Gaussians overlapping with the given pixel, and $T_i$ denotes the transmittance, defined as the product of opacity values of previous Gaussians overlapping the same pixel.

**Extract Multi-Granularity Scene Knowledge.** Upon preparing the attributes in the Gaussian primitives, we extract multi-granularity scene knowledge via foundation models. Different foundation models focus on different aspects of knowledge projection and granularity, as illustrated in Fig. 4. In this paper, we employ a set of foundation models $\mathbf{F} = \{\text{CLIP}, \text{SigLIP}, \text{DINOv2}, \text{LLaMA3}, \text{LLaMAv}, \text{SEEM}\}$, where LLaMAv is the vision-instruct version of LLaMA3. For each view, we extract foundation model embeddings, formally expressed as $\mathbf{F}(\mathbf{V}_*) = \mathbf{E} \in \mathbb{R}^{[h \times w, d]}$.

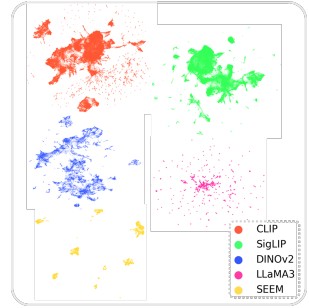

We implement specific algorithms for projecting LLaMA3 language embeddings and SEEM [52] visual prompts into pixel-level features. For LLaMA3, we first use SoM [43] and Semantic-SAM to extract language descriptions for each region. The language prompt of each region is represented as $\mathbf{T} \in \mathbb{R}^{[l_1, d]}$, where $l_1$ is the number of regions extracted by Semantic-SAM. For SEEM, we utilize visual prompts corresponding to each region, with visual prompts for each image represented as $\mathbf{O} \in \mathbb{R}^{[l_2, d]}$, where $l_2$ is the number of regions seg-

Figure 4: The UMAP visualization of model embedding manifolds reveals distinct shapes, reflecting different focus.

mented by SEEM. We then splat the features to the pixel level, duplicating the prompts within each mask region, resulting in $\mathbf{T}$ and $\mathbf{O}$ being indexed to the dimension of $\mathbb{R}^{[h \times w, d]}$.

After feature extraction, we obtain raw features ($\mathbf{R} \in \mathbb{R}^{[n, h \times w, d]}$) for the full Scene with $n$ views within each foundation model. These raw features span various granularities and knowledge spaces, providing a comprehensive multimodal (vision and language) understanding of the scene. In correlation to 3D Gaussian Splatting, the smallest granularity component is at the pixel level, with the most low-level knowledge projection being the RGB color value.

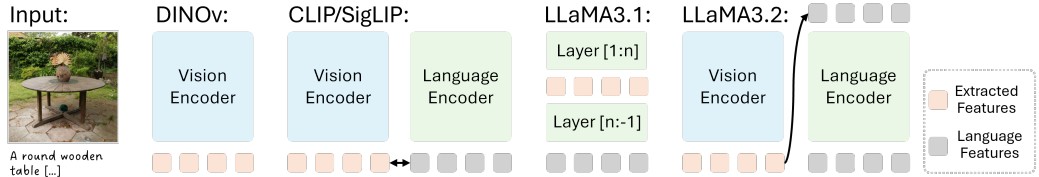

Figure 5: Illustration of patch-level visual embedding extraction their applications.

**Compress Scene Knowledge to Memory.** While the scene knowledge is extracted from foundation models $\mathbf{F}$ into raw feature space $\mathbf{R} \in \mathbb{R}^{[n,h\times w,d]}$, the dimensionality is too high for storage and rendering in each scene. Previous works such as F-3DGS [51] and F-Splat [26] address this issue through feature distillation. However, we observe two major problems with feature distillation: (a) The distilled feature experiences information loss compared to the original feature due to feature compression (usually 16 to 64 dims sampling linear projection to the original dimension $d$ that is usually 1000 dims or more). (b) The upsampled feature may have misalignments with the original knowledge space of the foundation model, making it difficult to be decoded by the original $\mathbf{F}$. To resolve these issues, we first flatten the raw features ($\mathbf{R}$) into $\mathbb{R}^{[n\times h\times w,d]}$, and then perform similarity reduction on the first dimension using Algorithm 1. The reduced raw feature represents the principal scene component (**PSC**), also named as memory bank, serving as the essential representation of the scene. The memory bank or **PSC** has dimensionality $\mathbb{R}^{[t,d]}$, where $t$ depends on the similarity threshold we set. The reduction is effective due to the presence of many duplicated features in neighboring spatial pixels within a view, or duplicated regions across views. We visualize the memory bank building process in Fig.3 a.

**Gaussian Memory Attention.** Given view-based principle queries $\mathbf{Q}_p^{\mathbf{V}_*} \in \mathbb{R}^{[H,W,n]}$ that is rasterized by gaussian primitives, and principle scene components $\mathbf{PSC} \in \mathbb{R}^{[l,d]}$, we perform Gaussian Memory Attention ($\mathbf{A}_{gm}$) to obtain the rendered feature in alignment with foundation models. With learnable random initialized memory projection $\mathbf{W}_m \in \mathbb{R}^{[n,d]}$, we formally define the Gaussian Memory Attention as follows:

$$\hat{\mathbf{R}} = \mathbf{A}_{gm}(\mathbf{Q}_p^{\mathbf{V}_*}) = \texttt{Softmax}(\mathbf{Q}_p^{\mathbf{V}_*} \times \mathbf{W}_m \times \mathbf{PSC}^T) \times \mathbf{PSC}. \tag{2}$$

This Gaussian memory attention links the $\mathbf{Q}_p$ with $\mathbf{PSC}$ and projects it into the corresponding foundation model knowledge space. The attention process is depicted in Fig. 3 b.

**Scene Rendering and Deployment.** Given the rendered features $\hat{\mathbf{R}}$ for each foundation model aligning with the corresponding foundation model space, we can link back to the powerful functions of foundation models. We expect that for models like `CLIP`, `SigLIP`, `SEEM`, the rendered feature can be directly used for vision language-based tasks such as retrieval and grounding. For generative-based models like `LLaMA3`, `LLaMAv`, we anticipate that the feature can be directly used in captioning or simple visual question answering tasks. Formally, we express this as $\mathcal{X} = \mathbf{F}_{\texttt{dec}}(\hat{\mathbf{R}})$.

## 4 EXPERIMENTS

### 4.1 EXPERIMENTAL SETUP

**Datasets.** To support extensive quantitative and qualitative evaluation, we perform experiments using several existing scene datasets [3; 18; 10] and collected a custom robot dataset (**M3**-Robot) using a quadruped robot and a drone. Specifically, we use *Garden* (an outdoor scene) from Mip-NeRF360 [3], *Train* from the Tank & Temples dataset [18], and *PlayRoom* as well as *DrJohnson* from the Deepblending dataset [10]. For the **M3**-Robot dataset, we collect images using two mobile robots. The *Table-Top* sequence is collected from a RealSense 405D camera mounted on the end effector of a Unitree Z1 robot arm on a Unitree B1 quadruped robot, where a human operator tele-operates the robot with a remote to obtain centripetal views of tabletop objects. Images in the *Geisel* sequence are collected by a tele-operated DJI Mini4-Pro drone. The collected images are processed by COLMAP [30] to obtain camera parameters and initialization.

**Memory across multiple Foundation Models.** The multi-modal memory mechanism allows **M3** to retain knowledge from many models, which differs from existing distillation-based methods that

| Dataset | Method | # Param | DINOv2 | | CLIP | | SigLIP | | SEEM | | LLaMA3 | | LLaMAv | |
|---|---|---|---|---|---|---|---|---|---|---|---|---|---|---|
| | | | Cosine↓ | L2↓ | Cosine↓ | L2↓ | Cosine↓ | L2↓ | Cosine↓ | L2↓ | Cosine↓ | L2↓ | Cosine↓ | L2↓ |
| Train | F-Splat [26] | 61M | 0.6833 | 1.9835 | 0.5998 | 0.4779 | 0.6346 | 0.7851 | 0.4269 | 11.72 | 0.5300 | 0.2900 | 0.7026 | 56.23 |
| | F-3DGS [51] | 61M | 0.3790 | 1.0108 | 0.3330 | 0.1540 | 0.3692 | 0.3328 | 0.1063 | 0.1034 | 0.4993 | 0.0150 | 0.6288 | 46.48 |
| | M3 | 35M | 0.5321 | 1.681 | 0.3140 | 0.2800 | 0.2811 | 0.5096 | 0.1389 | 0.2251 | 0.4401 | 0.0253 | 0.7069 | 53.43 |
| Garden | F-Splat [26] | 61M | 0.7328 | 1.9567 | 0.7005 | 1.3570 | 0.7247 | 0.8698 | 0.4224 | 9.4675 | 0.4944 | 0.3314 | 0.7443 | 60.83 |
| | F-3DGS [51] | 61M | 0.2295 | 0.6033 | 0.2105 | 0.0945 | 0.2697 | 0.2585 | 0.1071 | 0.1424 | 0.4139 | 0.0141 | 0.4913 | 43.08 |
| | M3 | 35M | 0.5701 | 1.7279 | 0.3168 | 0.2876 | 0.2927 | 0.0004 | 0.1839 | 0.3469 | 0.3387 | 0.0217 | 0.7235 | 58.04 |
| Drjohnson | F-Splat [26] | 61M | 0.8107 | 2.0333 | 0.6689 | 0.7877 | 0.6826 | 0.7744 | 0.4650 | 10.411 | 0.3757 | 0.0145 | 0.8184 | 54.82 |
| | F-3DGS [51] | 61M | 0.4190 | 1.1279 | 0.3344 | 0.1537 | 0.3846 | 0.3552 | 0.1693 | 0.2169 | 0.3853 | 0.0150 | 0.6669 | 47.35 |
| | M3 | 35M | 0.5878 | 1.7553 | 0.3435 | 0.2924 | 0.2975 | 0.5366 | 0.2456 | 0.4179 | 0.3175 | 0.0226 | 0.7224 | 52.68 |
| Playroom | F-Splat [26] | 61M | 0.7956 | 1.9640 | 0.6458 | 0.7808 | 0.6839 | 0.7678 | 0.4745 | 10.873 | 0.3915 | 0.0136 | 0.8185 | 59.42 |
| | F-3DGS [51] | 61M | 0.4867 | 1.2193 | 0.3813 | 0.1726 | 0.4571 | 0.4094 | 0.1714 | 0.2103 | 0.3987 | 0.0139 | 0.6922 | 52.50 |
| | M3 | 35M | 0.6074 | 1.7545 | 0.3260 | 0.2987 | 0.2951 | 0.5623 | 0.2560 | 0.4584 | 0.3555 | 0.0241 | 0.7288 | 57.38 |

Table 1: Feature Distance in comparison with distillation methods that use similar or higher budgets across datasets and foundation models.

| Dataset | Method | #Param | CLIP | | | | SigLIP | | | | | |
|---|---|---|---|---|---|---|---|---|---|---|---|---|
| | | | mIoU | cIoU | AP50 | AP60 | I2T@1 | I2T@5 | I2T10 | T2I@1 | T2I@5 | T2I10 |
| Train | **Ground Truth** | - | 25.3 | 26.3 | 14.7 | 3.3 | 81.5 | 97.3 | 100.0 | 71.0 | 89.4 | 92.1 |
| | F-3DGS [51] | 61M | 24.2 | 24.3 | 16.3 | 7.1 | 2.6 | 13.2 | 28.9 | 0.0 | 2.6 | 18.4 |
| | **M3** | **35M** | **25.4** | **26.5** | **19.6** | **12.5** | **55.2** | **84.2** | **92.1** | **52.6** | **84.2** | **92.1** |
| Playroom | **Ground Truth** | - | 25.6 | 24.2 | 9.6 | 3.0 | 96.5 | 100.0 | 100.0 | 62.0 | 96.5 | 100.0 |
| | F-3DGS [51] | 61M | 23.8 | 21.4 | 11.9 | 3.0 | 79.3 | 96.6 | 96.6 | 31.0 | 79.3 | 89.7 |
| | **M3** | **35M** | **23.1** | **23.1** | **11.9** | **5.9** | **72.4** | **96.6** | **100.0** | **41.3** | **65.5** | **68.9** |
| Geisel | **Ground Truth** | - | 19.5 | 21.4 | 5.3 | 0.0 | 100.0 | 100.0 | 100.0 | 60.0 | 85.7 | 91.4 |
| | F-3DGS [51] | 61M | 19.0 | 20.4 | 14.1 | 1.2 | 45.7 | 94.3 | 100.0 | 0.0 | 20.0 | 34.3 |
| | **M3** | **35M** | **21.8** | **23.5** | **16.5** | **11.8** | **100.0** | **100.0** | **100.0** | **71.4** | **85.7** | **94.2** |

Table 2: Feature/RGB metrics for all foundation models and scene.

only distill a few (2-3) models. Specifically, as provided in Sec. 3.3, we employ 6 foundation models to resemble human memory of different aspects. Each model has different granularity and focus of different semantics: image-level vision-language understanding via CLIP [28] as well as SigLIP [47]; pixel-level semantic understanding via SEEM [52]; self-supervised structural feature via DINOv2 [24]; and LLaMA3.1/3.2v [6] for multi-modal understanding and reasoning.

In Fig. 5, we provide a comprehensive illustration of how we extract features from foundation models. The extracted features are marked in orange in alignment with language representations optionally or continued to be the input of the language Encoder.

**Loss Computation.** For each input image, we extract patch-level embeddings from the aforementioned models. Previous methods [26; 51] compute the patch-wise distance loss on the rendered features, this not only has a high volume of GPU memory consumption that hinders parallel training for all the foundation models but also creates artifacts when downsampling the feature. In compensate, we use point-based loss, where we sample 2000 points ranging from both predict and ground truth features for distance loss computation. This largely reduces the computation overhead for training, as shown in Table. 1.

**Low-level Evaluation Metrics.** To systematically evaluate multi-modal memory, we use evaluation metrics ranging from low/pixel-level to high-level downstream tasks. In particular, the low-level evaluation metrics evaluate pixel-level image quality. For rendered image quality on evaluation views (views not provided in training), we use common metrics (PSNR, SSIM, and LPIPS [49]) as Kerbl et al. [16]. For feature quality, we report cosine and L2 distance.

**High-level Evaluation Metrics.** High-level evaluation metrics, different from low-level ones, focus on evaluating downstream tasks of features. For discriminative models [28; 47; 52], we will report commonly used metrics such as mIoU (mean Intersection over Union), cIoU (complete Intersection over Union), and AP (Average Precision). For retrieval, we will use IR@1 (Information Retrieval at rank 1) and TR@1 (Text Retrieval at rank 1).

## 4.2 QUANTITATIVE RESULTS

**Baseline Implementation** For quantitative experiments, we compare **M3** with two recent distillation-based feature GS methods [26; 51]. For fair comparisons, we train all the methods in approximately 30,000 iterations (29,993 iterations for **M3** due to last-batch data loader roundoffs). The reference training features are identical for different methods. For distillation-based methods, we follow F-Splat [26] to render a latent feature and then decode the latent features to the embedding

| Dataset | Method | RGB PSNR↑ | Time min. | CLIP Cosine↓ | L2↓ | SigLIP Cosine↓ | L2↓ | DINOv2 Cosine↓ | L2↓ | SEEM Cosine↓ | L2↓ | LLaMA3 Cosine↓ | L2↓ | LLaMAv Cosine↓ | L2↓ |
|---|---|---|---|---|---|---|---|---|---|---|---|---|---|---|---|
| Tabletop | +CLIP | 21.91 | ∼6 | 0.3100 | 0.2956 | - | - | - | - | - | - | - | - | - | - |
| | +SigLIP | 21.84 | ∼10 | 0.3100 | 0.2956 | 0.3122 | 0.0005 | - | - | - | - | - | - | - | - |
| | +DINOv2 | 21.79 | ∼15 | 0.3101 | 0.2956 | 0.3123 | 0.0005 | 0.5161 | 1.6057 | - | - | - | - | - | - |
| | +SEEM | 21.93 | ∼20 | 0.3101 | 0.2956 | 0.3123 | 0.0005 | 0.5156 | 1.6048 | 0.0472 | 0.1013 | - | - | - | - |
| | +LLaMA3 | 21.97 | ∼30 | 0.3101 | 0.2956 | 0.3122 | 0.0005 | 0.5160 | 1.6056 | 0.0472 | 0.1012 | 0.3628 | 0.0246 | - | - |
| | +LLaMAv (All) | 21.96 | ∼45 | 0.3100 | 0.2956 | 0.3122 | 0.0005 | 0.5157 | 1.6049 | 0.0472 | 0.1013 | 0.3628 | 0.0246 | 0.7262 | 59.92 |

Table 3: Ablation on the number of foundation models in **M3**.

| Degree | # Params | Iteration | CLIP Cosine↓ | L2↓ | mIoU | AP50 | SigLIP Cosine↓ | L2↓ | mIoU | AP50 | DINOv2 Cosine↓ | L2↓ | SEEM Cosine↓ | L2↓ | LLaMA3 Cosine↓ | L2↓ |
|---|---|---|---|---|---|---|---|---|---|---|---|---|---|---|---|---|
| 8x6=48 | 14.8M | 30k | 0.3256 | 0.2880 | 25.4 | 19.6 | 0.2913 | 0.5239 | 19.4 | 2.1 | 0.5755 | 1.7664 | 0.1672 | 0.2749 | 0.4504 | 0.0264 |
| | | 7k | 0.3290 | 0.2900 | 25.3 | 14.6 | 0.2938 | 0.5277 | 21.8 | 4.8 | 0.5845 | 1.7835 | 0.2058 | 0.3463 | 0.4517 | 0.0265 |
| 16x6=96 | 21.5M | 30k | 0.3140 | 0.2800 | 25.7 | 19.0 | 0.2866 | 0.5172 | 24.3 | 10.3 | 0.5535 | 1.7239 | 0.1388 | 0.2247 | 0.4480 | 0.0261 |
| | | 7k | 0.3206 | 0.2842 | 25.3 | 20.6 | 0.2903 | 0.5227 | 23.2 | 8.1 | 0.5677 | 1.7513 | 0.1828 | 0.3056 | 0.4504 | 0.0263 |
| 32x6=192 | 34.8M | 30k | 0.3043 | 0.2735 | 26.7 | 22.8 | 0.2814 | 0.5094 | 25.7 | 11.9 | 0.5318 | 1.6807 | 0.0972 | 0.1553 | 0.4401 | 0.0253 |
| | | 7k | 0.3132 | 0.2792 | 26.2 | 21.1 | 0.2866 | 0.5172 | 25.5 | 11.4 | 0.5515 | 1.7198 | 0.1269 | 0.2139 | 0.4436 | 0.0256 |
| 64x6=384 | 61.4M | 30k | 0.2917 | 0.2650 | 28.4 | 23.9 | 0.2721 | 0.4957 | 28.5 | 13.5 | 0.5099 | 1.6358 | 0.0855 | 0.1321 | 0.4278 | 0.0241 |
| | | 7k | 0.3049 | 0.2734 | 28.1 | 23.9 | 0.2802 | 0.5079 | 27.8 | 13.5 | 0.5350 | 1.6870 | 0.1012 | 0.1676 | 0.4348 | 0.0248 |

Table 4: Ablation on the dimensions and distillation for each foundation model.

space of reference features with a multi-head MLP. For all methods, the optimization of both latent features/memory and decoders is trained from scratch for each scene.

**Low-Level Results.** We report the main quantitative results in Tab. 1, where the average training time and the auxiliary low-level metrics are reported. Our method, **M3**, outperforms F-Splat while reducing significantly compute than F-3DGS. `SEEM` and `LLaMA3` features extraction failed on F-Splat, which we assume was mainly due to the ground truth feature extraction procedure, where duplication was performed to each segmentation to get pixel-level features.

**Downstream Results.** The downstream evaluation results of grounding and retrieval are shown in Table. 2. The ground truth grounding dataset of Train, Playroom, and Geisel is generated by SoM [] with semantic-SAM for mask labels and GPT4-o [] to generate the caption. Example data are shown in the Appendix. We evaluate all the images in the validation sets of the three datasets. The grounding results clearly show that **M3** is better than F-3DGS with half of the parameters, the gap is non-trivial especially when looking at the AP50/AP60 columns. In addition to grounding results, we also evaluate **M3** on image text retrieval, similar to grounding we use GPT4-o to generate ground truth data for three datasets. The example data are also shown in the appendix. Compared to grounding performance, **M3** performs much better than F-3DGS on retrieval results. For image text retrieval, the positive example is the evaluation image, and the negative pairs are generated by COCO datasets. We believe the large gap in the retrieval results is taken from Gaussian memory attention, where the rendered features are aligned with the original foundation model much better. When we find the correct embeddings in the dataset, this benefit gets enlarged.

**Ablation Results.** Table. 3 shows the ablation of the number of foundation models involved in **M3**. We gradually added foundation models from simpler single-modal models to more advanced multi-modal models. While maintaining a very efficient training time, our method has independent results from different foundation models. Our implementation is based on Grendel-GS, where the training procedure is efficiently paralleled. In Table. 4, we ablate the computation budget on training **M3** in the balance of memory footprint, training iterations, and performance. The table clearly shows that increasing the number degree will generally improve the performance on all metrics. While having 16 degrees for each foundation model is enough to obtain a reasonable performance. That is what the number is reported in the paper. In addition, increasing training iteration will generally increase the performance, while 1/4 of the training budget (7k) would usually get a reasonable performance.

## 4.3 QUALITATIVE RESULTS.

**M3** consistently demonstrates superior performance across diverse datasets as shown in Fig. 6, by effectively preserving fine-grained details and ensuring smooth, coherent feature representations. The method excels at retaining intricate details such as the textures of chairs and the fine features of books, highlighting its ability to capture micro-level information. This clear layering contributes to rich semantic understanding within the scenes.

Furthermore, **M3** handles overlapping objects exceptionally well, as evident in the *Playroom* dataset, where complex arrangements are rendered with accurate structural information. The outputs from various foundation models are consistently high-quality, each retaining spatial structures and semantic information at different granularities. This demonstrates **M3**'s capability to capture both

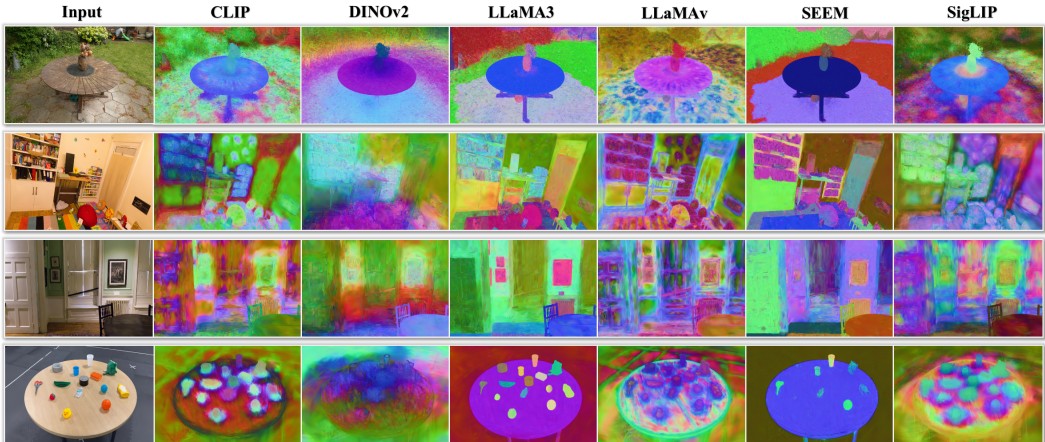

Figure 6: Qualitative results across datasets using **M3**. The figure showcases the consistent performance of the **M3** across various datasets (*Garden*, *Playroom*, *Drjohnson*, *Table-top*).

low-level spatial details and high-level semantic concepts, making it highly effective for tasks that require comprehensive scene understanding.

### 4.4 DEMONSTRATION RESULTS.

We also deployed **M3** on a quadruped robot platform to demonstrate the potential real world applications of our model. In this experiment, we first tele-operate the robot to scan the table by taking a centripetal video with onboard camera. After memorizing the scene with **M3**, the robot is able to locate and grasp any object with text query on decoded CLIP feature. With known robot pose through LiDAR, we are able to render any camera pose $_{c_0}T_{c_t}$ with:

$$_{c_0}T_{c_t} = \ _{c_0}T_{e_0} \times \ _{e_0}T_{b_0} \times \ _{b_0}T_{l_0} \times \ _{l_0}T_w \times \ _wT_{l_t} \times \ _{l_t}T_{b_t} \times \ _{b_t}T_{e_t} \times \ _{e_t}T_{c_t} = \ _{l_0}T_w \times \ _wT_{l_t}, \quad (3)$$

where $c, e, b, l$ and $w$ refer to `camera`, `end effector`, `arm base`, `lidar`, and `world` respectively. Note that to align with COLMAP coordinates, the camera pose needs modifying with $_{COLMAP_0}T_{c_0} \times \ _{c_0}T_{c_t}$.

We tested with the query "yellow bath duck" on the decoded CLIP feature, and as shown in Fig. 7, the rubber duck is highlighted in red. The robot can then locate the 3D position of the targeted object with depth information from its depth camera and perform a grasping task.

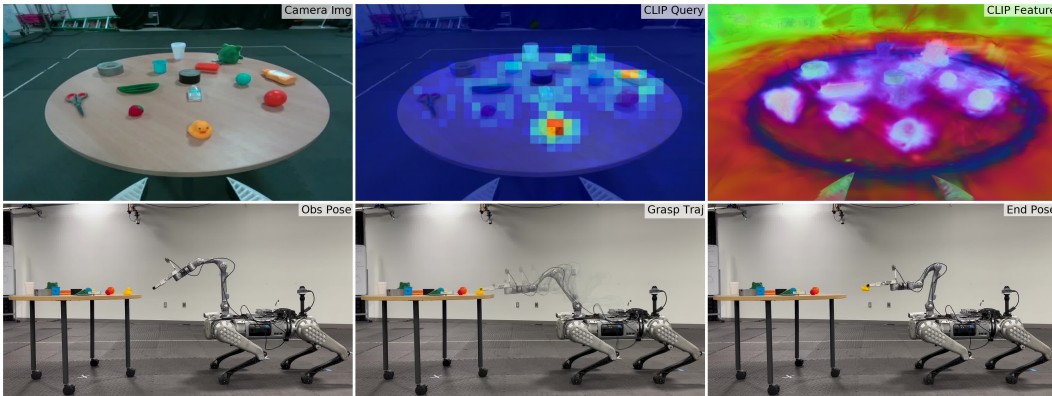

Figure 7: Real robot deployment.

**Conclusion.** This paper introduces **M3**, a novel approach combining foundation models with Gaussian Splatting to create a spatial multimodal memory resembling human memory. **M3** demonstrates superior downstream task accuracy with reduced training costs and shows practical utility when deployed on a real robot. One interesting future direction is to design a reasoning module that is capable of directly operating on the optimized memory bank, which we leave to future study.

**Acknowledgement.** This work was supported, in part, by NSF CAREER Award IIS-2240014, and NSF CCF-2112665 (TILOS). This research project has benefitted from the Microsoft Accelerate Foundation Models Research (AFMR) grant program.

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
