## A   CLARIFICATION

**Terminology and Notations.**   In order to minimize confusion of the terminologies, here we give comprehensive description of visual granularity, knowledge space, and principal scene components with figure illustration for each concept.

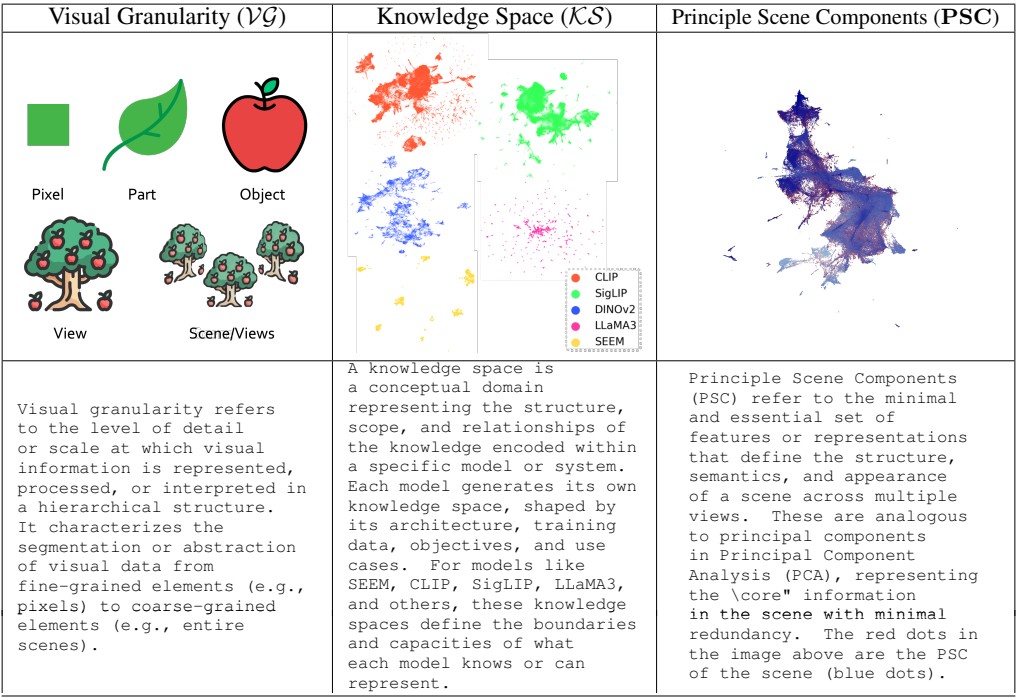

| Visual Granularity ($\mathcal{VG}$) | Knowledge Space ($\mathcal{KS}$) | Principle Scene Components (**PSC**) |
|---|---|---|
| Visual granularity refers to the level of detail or scale at which visual information is represented, processed, or interpreted in a hierarchical structure. It characterizes the segmentation or abstraction of visual data from fine-grained elements (e.g., pixels) to coarse-grained elements (e.g., entire scenes). | A knowledge space is a conceptual domain representing the structure, scope, and relationships of the knowledge encoded within a specific model or system. Each model generates its own knowledge space, shaped by its architecture, training data, objectives, and use cases. For models like SEEM, CLIP, SigLIP, LLaMA3, and others, these knowledge spaces define the boundaries and capacities of what each model knows or can represent. | Principle Scene Components (PSC) refer to the minimal and essential set of features or representations that define the structure, semantics, and appearance of a scene across multiple views. These are analogous to principal components in Principal Component Analysis (PCA), representing the \core" information in the scene with minimal redundancy. The red dots in the image above are the PSC of the scene (blue dots). |

Below are additional notations and explanations we used in the paper:

**Principle Query** ($\mathbf{Q}_p$): A variable within Gaussian primitives, designed to encode low-rank embeddings.

**Scene** ($\mathbf{V}$): A 3D environment observable from multiple viewpoints.

**View** ($\mathbf{V}_*$): A single perspective or projection of the 3D scene.

**Foundation Models** ($\mathbf{F}$):** Large vision-language models that map views into structured knowledge spaces.

**Embeddings** ($\mathbf{E}$): Outputs generated by foundation models, representing data in their respective knowledge spaces.

**Raw Features** ($\mathbf{R}$): A comprehensive collection of embeddings produced by foundation models, encompassing the full knowledge space.

**Rendered Feature** ($\hat{\mathbf{R}}$): Features derived through Gaussian splatting, computed using Gaussian memory attention.

**Gaussian Memory Attention.**   Gaussian Memory Attention, as defined in the main paper, is the procedure to render the raw feature from principal query of a single view:

$$\hat{\mathbf{R}} = \mathbf{A}_{gm}(\mathbf{Q}_p^{\mathbf{V}_*}) = \texttt{Softmax}(\mathbf{Q}_p^{\mathbf{V}_*} \times \mathbf{W}_m \times \mathbf{PSC}^T) \times \mathbf{PSC}. \qquad (4)$$

The high level logic of Gaussian Memory Attention is to first project the principal query ($\mathbf{Q}_p^{\mathbf{V}_*}$), which the the compressed representation of **PSC** into its original dimensionality. Then we compute the similarity of the up-sampled principal queries ($\mathbf{Q}_p^{\mathbf{V}_*} \times \mathbf{W}_m$) with principal scene components. Finally, according to the similarity score, we do a weighted sum of the principal scene components, the resulted feature is the final rendered feature in-aligned with foundation model features.

In the figure below, we show the visualization of principal query, psc, raw feature and the final render feature. **All the images, including the circles (●, ●, ●, ●) are directly draw by algorithm.** We will explain how each component is drawn in details:

**Principal Query:** Given an image with size $[h, w]$, the rendered principal queries of the given view is $[c, h, w]$. We compute the umap of the principal query and downsample the feature dimension to 3, and visualize umap feature in rgb format via colormap. We overlay the original rgb image and the visualization of rendered pricipal components. Finally, in order to use visualization to proof-of-concept of Gaussian Memory Attention, we sample four principal queries on location $[0.25, 0.25], [0.25, 0.75], [0.75, 0.25], [0.75, 0.75]$. And those four points are corresponding to the circle drawn in the image with ●, ●, ●, ●.

**Principal Scene Component** (best viewed with zoom-in) **:** The principal scene components are the subset of raw feature. In the second column, we visualize the umap down-sampled principal scene component (●) and around 1/20 original raw feature (●). The top-5 PSC components to the corresponding principal queries are represented with ●, ●, ●, ● again. In this way, we could know where the original four pixel of principal query falls in the PSC space.

**Principal Scene Component** (best viewed with zoom-in) **:** This part is the most important in the table. It traces back to the original feature location for the top-1 PCA component in the second column. The circles ●, ●, ●, ● are draw with the center of the traced pixel in the feature map. It clearly shows that the train, sky, ground, stars are clearly correlated to the correct PSC component that is part of the original raw features.

**Render Feature:** Finally we clearly show the final rendered feature after gaussian memory attention in the last column.

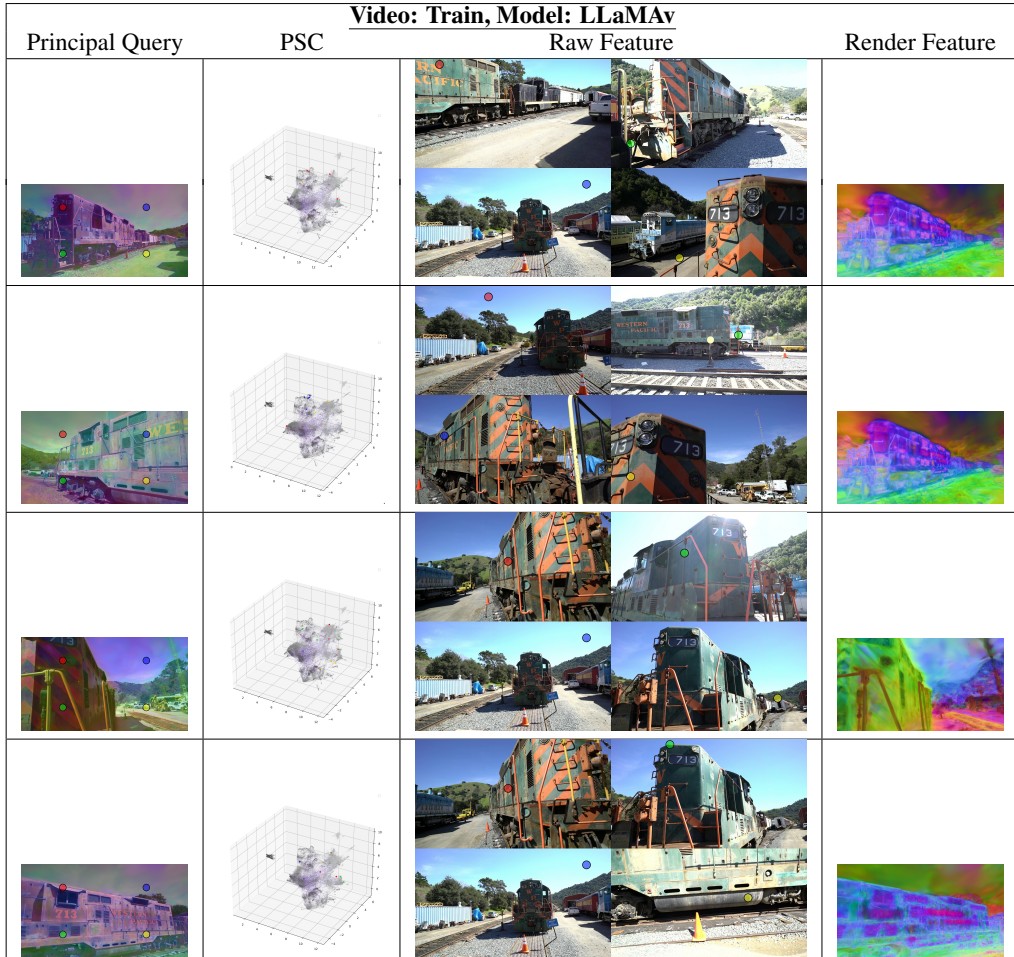

# B  M3 LMM Benchmark

**Grounding.** We create the LMM evaluation benchmark on grounding using SoM [43] and Semantic-SAM [20]. The pipeline first uses Semantic-SAM to label the marked image and then uses SoM and GPT4-o to label the marked region with proper text. Below we show examples of the datasets Train, Geisel, and Garden.

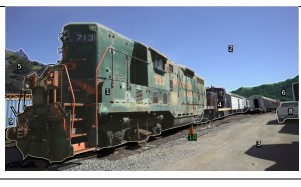
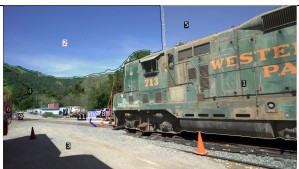
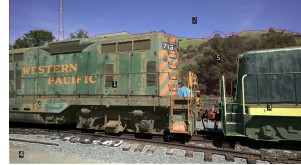

| | | |
|---|---|---|
| 1.Green locomotive with number 713. 2.Clear blue sky. 3.Dirt and gravel ground. 4.Rusty train cars in a row. 5.Green hills in the background. | 1.Green train engine with '713' and 'WESTE' visible. 2.Blue sky with some clouds. 3.Gravel path or road. 4.Green hills or mountain range. 5.Tall pole or antenna. | 1.Green locomotive with 'Western Pacific' written on the side. 2.Blue sky with a few clouds. 3.Part of a dark-colored train car. 4.Gravel and railroad tracks. 5.Green hills and trees in the background. |

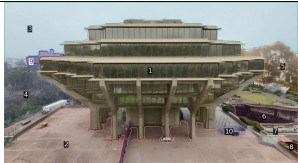
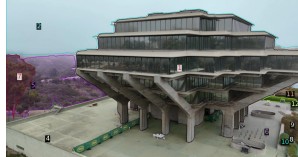
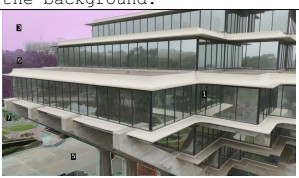

| | | |
|---|---|---|
| 1.Large, multi-level building with extensive glass windows. 2.Open concrete plaza area in front of the building. 3.Clear sky above the building. 4.Trees and greenery surrounding the area. 5.Hillside with sparse vegetation. | 1.Modern library architecture with large glass windows. 2.Clear blue sky in the background. 5.Distant mountain range visible in the horizon. 7.Tall trees with dense foliage. 9.Green grass area near the building. | 1.The structure of the library with large glass windows. 3.The sky above the building. 5.The concrete land area surrounding the library. 6.A group of trees in the background. 7.A grassy area near the library. |

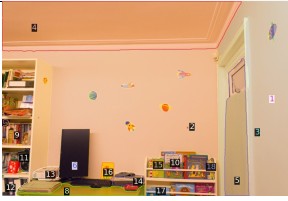
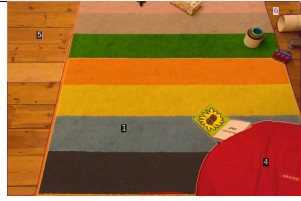
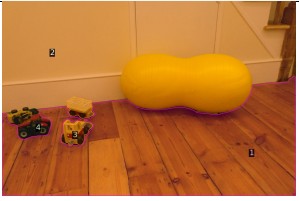

| | | |
|---|---|---|
| 1.A door on the right side of the image. 2.A wall with space-themed stickers. 4.The ceiling with decorative molding. 6.A monitor on the table. 9.A bookshelf filled with various items. | 1.A colorful striped rug on the floor. 2.A red bean bag chair. 5.A wooden floor beside the rug. 6.A set of stacked toy cups on the rug. | 1.Wooden floorboards covering the ground. 2.White wall with a baseboard. 3.Small toy truck on the floor. 4.Toy tractor placed next to the toy truck. |

**Retrieval & Captioning.** Similar to grounding, we also use SoM [43] and Semantic-SAM [20] labeled images to generate the description for the evaluation dataset including Train, Geisel, Garden, Drjohnson, and Playroom. Below, we show examples in Drjohnson and Garden to show variants with the grounding examples.

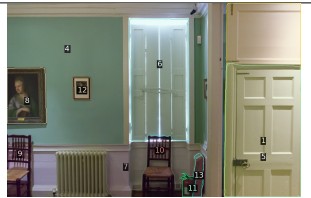

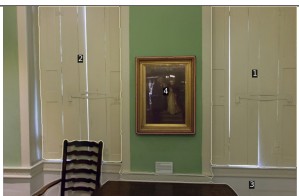

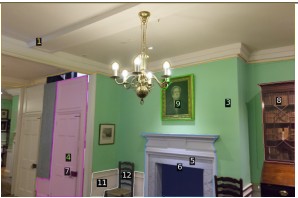

**Caption:** A room features chairs, artwork on the walls, and a window with shutters.
**Short Caption:** A room with chairs and artwork.
**Long Caption:** The image shows an interior room with light green walls. There are two wooden chairs positioned beneath artwork, a radiator along the wall, and a window with closed shutters allowing light to stream in. A wooden door with a handle is on the right side.

**Caption:** A room featuring a framed painting between two shuttered windows.
**Short Caption:** Room with a painting and shutters.
**Long Caption:** The image shows a room with two large shuttered windows on either side of a framed painting hung on a green wall. Below the painting is a chair and a table, creating a classic and elegant atmosphere.

**Caption:** A well-decorated room featuring a chandelier, fireplace, and framed artwork.
**Short Caption:** Elegant room with a chandelier and fireplace.
**Long Caption:** The room is elegantly decorated with a classic chandelier hanging from the ceiling, a white fireplace with a dark opening, and a framed portrait on the green wall. Bookshelves filled with books are visible on the right, while a couple of chairs are placed near the fireplace. The room has an inviting, traditional atmosphere.

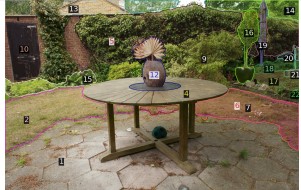

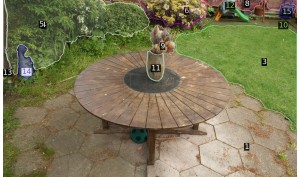

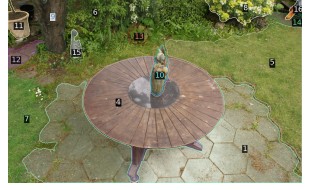

**Caption:** A round wooden table sits in a garden, featuring a vase with decorative items as a centerpiece.
**Short Caption:** Garden table with a vase centerpiece.
**Long Caption:** In a quaint garden setting, a round wooden table is centrally placed on a tiled patio. The table's focal point is a vase filled with decorative items, adding a touch of elegance. Surrounding the table are well-kept grass, various plants, and a backdrop of a brick wall covered in ivy, creating a serene outdoor atmosphere.

**Caption:** A wooden table is placed in a garden surrounded by grass and plants.
**Short Caption:** Garden scene with a wooden table.
**Long Caption:** The image depicts a backyard garden scene featuring a wooden circular table set on a stone patio. The garden is lush with greenery, including various bushes and plants. A vase with decorative elements sits atop the table, and there's a small soccer ball underneath. In the background, a mix of garden furniture and toys is visible, surrounded by well-maintained grass and foliage.

**Caption:** A round wooden table sits in a garden surrounded by greenery.
**Short Caption:** Garden with a round table.
**Long Caption:** A round wooden table is at the center of a garden scene. The table is on a hexagonal stone patio and is surrounded by various plants and grass. A small statue sits in the middle of the table, adding a decorative touch. The lush garden includes a mix of potted plants, bushes, and a tree, creating a serene outdoor space.

## C  QUALITATIVE RESULTS

**Real Robot Experiments.**    In addition to the main paper. We conduct additional real robot experiments on (1) Part-level understanding. (2) Multi-Scene Scenario. (3) Long-Horizon Task. Below we show the command and demo video frames for each task. Note, for the multi-scene task, we use CLIP and SigLIP for grounding different scenes to achieve the multi-foundation model scenario.

(1) Pick up the screwdriver by its handle and place it on the plate.

(2) Pick up the sponge and place it into the sink.

(3) Pick up the pineapple from the kitchen set and place it on to the table.

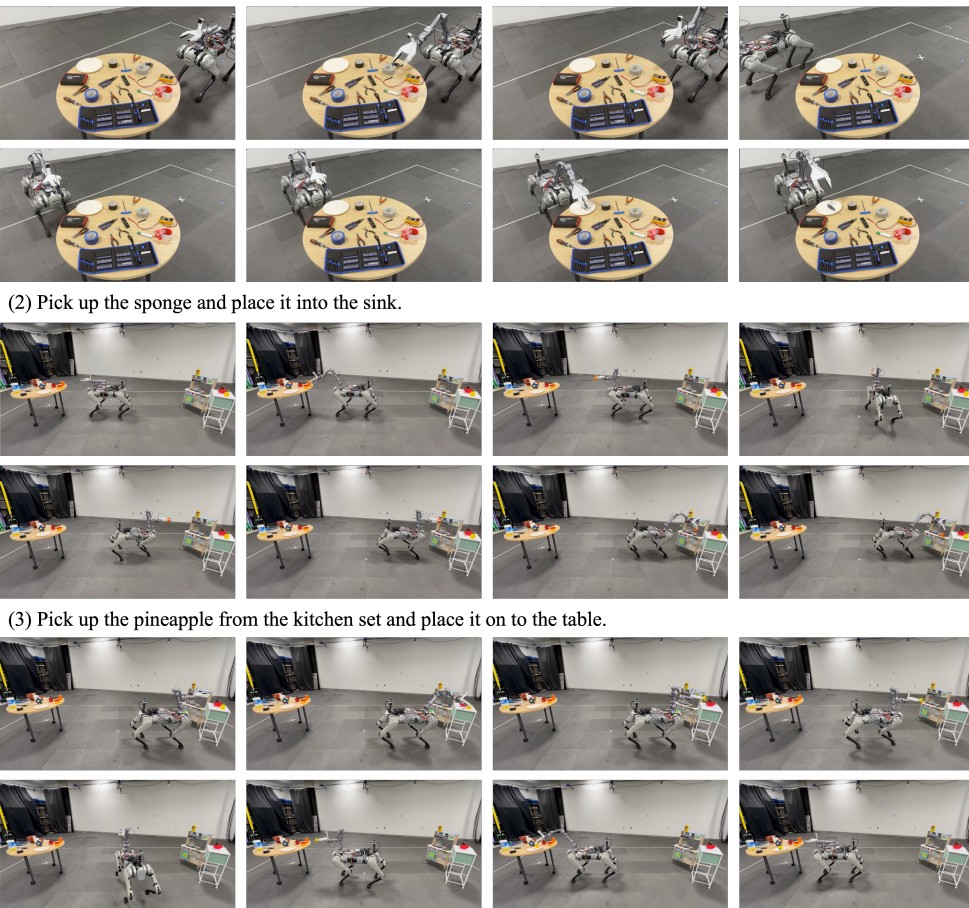

Figure 8: Real robot deployment on part-level understanding, multi-scene and long-horizon tasks.

**Feature Comparison.** Here we visualize the feature PCA comparison between **M3** and F-3DGS. Noted that different from the original paper of F-3DGS that uses semantic clustering for VLM features, in our work, we train both **M3** and F-3DGS for the original VLM features for fair comparison. Looking at the visualization below, we can observe that while numerical performance is not too far away between these two methods, the feature quality and continuity of F-3DGS are much lower than **M3**. We show features from both CLIP and SigLIP models on train and Geisel datasets.

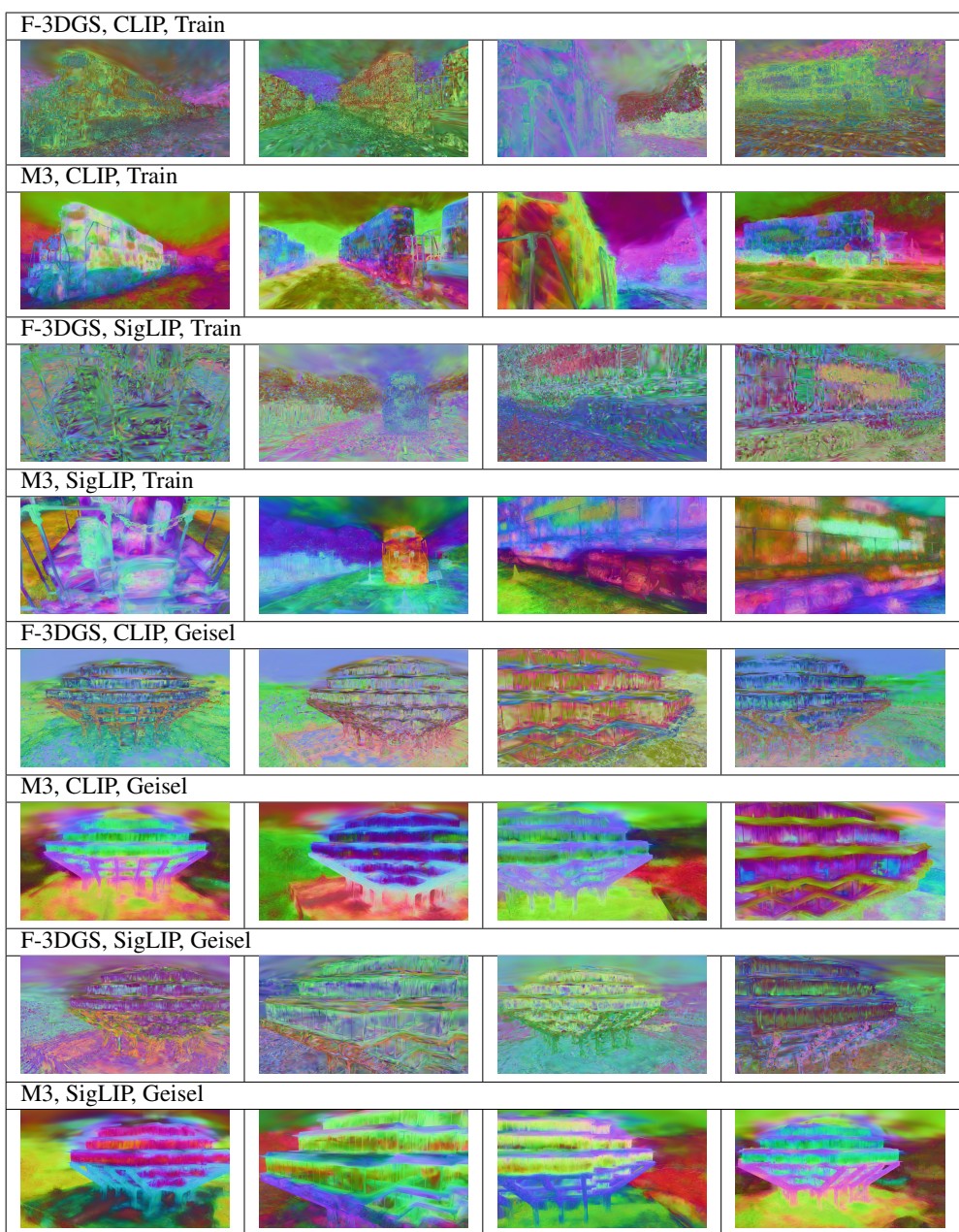

**Knowledge Space ($\mathcal{KS}$) Visualization.** We visualize the knowledge space across datasets and foundation models. As shown in the figure below, we visualize both the Tabletop and Train dataset. The knowledge space manifold was built by all the feature pixels for each foundation model in the video, the blue point is the raw feature, and the red point is the principal component, the first and second rows are multi-view visualizations of the feature manifold. We could interestingly observe that the feature manifold pattern is different across foundation models, and especially the LLaMAv feature is the most diverse and continuous. This is actually interestingly aligned with the feature visualization of Fig.6 in the main paper.

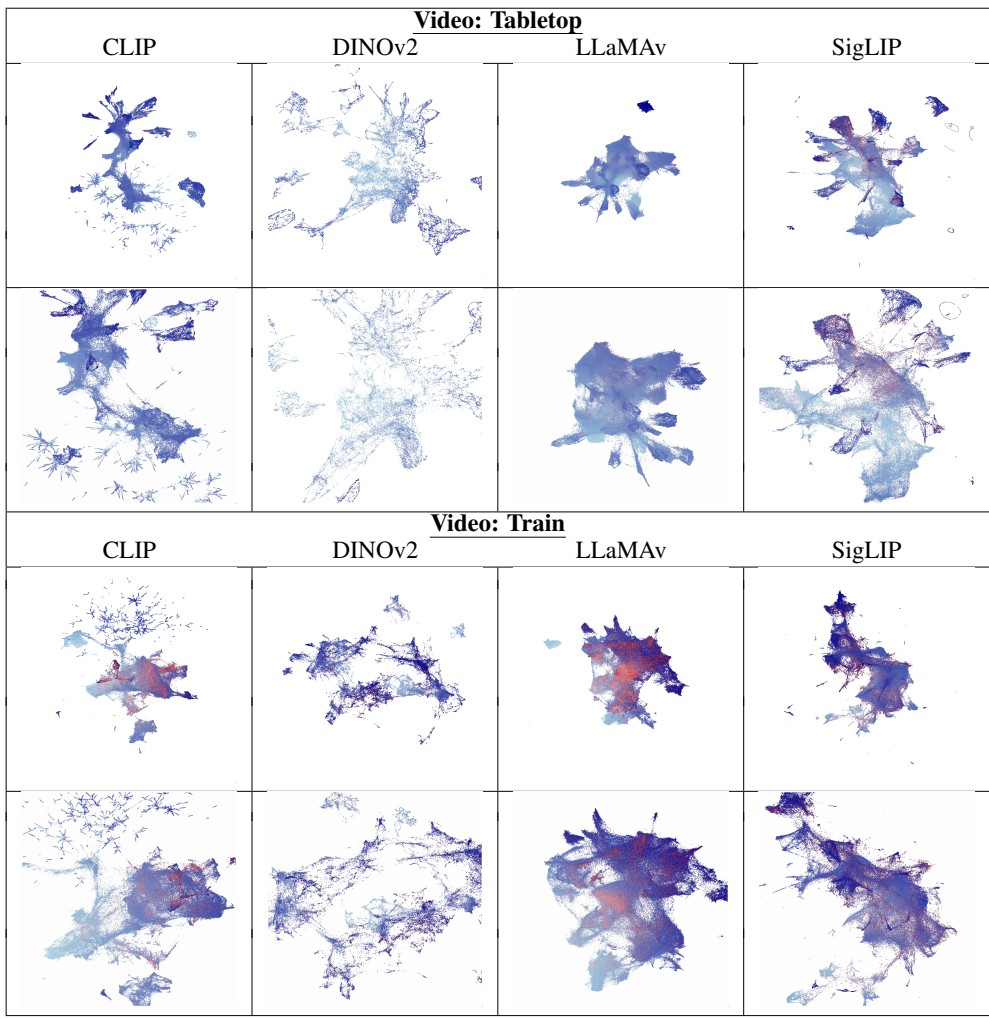