# OpenReview forum: "3D-SPATIAL MULTIMODAL MEMORY"
_ICLR.cc/2025/Conference — ICLR 2025 Poster_

### Official Review · Reviewer_2XXj · 2024-10-31

**Soundness:** 3
**Presentation:** 3
**Contribution:** 2
**Rating:** 6
**Confidence:** 2

**Summary:**

This paper introduces 3D Spatial MultiModal Memory (M3), a video memory system aimed at retaining visual perception information over medium temporal and spatial scales. M3 employs 3D Gaussian splatting and incorporates multiple foundation models to create a multimodal memory capable of spanning visual details and encompassing a broad knowledge range. The authors address two primary limitations of prior feature splatting methods: high computational demands for storing dense features in each Gaussian primitive and misalignments with foundation model knowledge. M3 innovatively introduces principal scene components and Gaussian memory attention mechanisms to improve the efficiency of storage, training, and inference within the Gaussian Splatting Model. Evaluated across various foundation models, including vision-language and self-supervised models, M3 demonstrates robust performance, with successful deployment on a quadruped robot for grasping tasks, showcasing its practical viability in real-world applications.

**Strengths:**

The idea is interesting and novel, and this direction is promising for diverse robotic tasks.
The method design makes sense. The figures and algorithm flowchart make the paper easy to understand.
The authors provide comprehensive coparisons in evaluation on M3 features (table 1, 2, 3, 4).

**Weaknesses:**

My concerns mostly lie on real robot experiments. I think one of the advantage of such feature field (also claimed in paper) is for various downstream robotics manipulation tasks. While, this paper shows a simple example in grasping a duck on the table. It could make this paper much stronger if the authors can demonstrate the promising potential of this study in various practical downstream tasks.
Some example downstream tasks:
1. Geometry-based or semantic-part-based tasks using DINO or diffusion features, such as geometry-based object retrieval/grasping[1, 2], planning objects to the goal configuration[3].
2. Language-guided tasks using CLIP feature, such as like "pick up the red cube and place it on the blue book." (example references: [4, 5])
3. A multi-step task that combines different features, such as "navigate to the kitchen, locate the mug with a cat design, and bring it to the living room table." These examples would showcase how M3 can leverage different types of features for more complex robotic applications.

Some typos: for example, in figure 5, dinov->dinov2. In line 350&351, there are missing parts after `as shown in'.

[1] SparseDFF: Sparse-View Feature Distillation for One-Shot Dexterous Manipulation. ICLR 2024.

[2] Robo-ABC icon : Affordance Generalization Beyond Categories via Semantic Correspondence for Robot Manipulation. ECCV 2024.

[3] D^3Fields: Dynamic 3D Descriptor Fields for Zero-Shot Generalizable Rearrangement. CoRL 2024.

[4] Distilled Feature Fields Enable Few-Shot Language-Guided Manipulation. CoRL 2023.

[5] NaturalVLM: Leveraging Fine-grained Natural Language for Affordance-Guided Visual Manipulation. RA-L 2024.

**Questions:**

Please see weakness.

---

> ### Author Response · Authors · 2024-11-28
> **Response to Reviewer 2XXj (1/2)**
>
> Rebuttal Resources to Reviewer `2XXj`: [[rebuttal pdf: 2XXj]](https://drive.google.com/file/d/1KYrD-ypXtmyX7IQ5mrGoo2fRgEhdY3ou/view?usp=sharing), [[video: pickup sponge]](https://drive.google.com/file/d/1v5JtroD3jYpvyWnoFNU5Vcm3Q3bcrA7r/view?usp=sharing), [[video: pickup pineapple]](https://drive.google.com/file/d/1mWjn2_ssM1MvFZtIY0wzOjR28HiUNFBf/view?usp=sharing), [[video: pickup screwdriver]](https://drive.google.com/file/d/1Oimra-s_nwakyoYtJbekJTb_mj4FNKJA/view?usp=sharing)
>
> ---
>
> We sincerely appreciate reviewer `2XXj` for the positive comments on our contributions, especially for the potential applications in diverse robotic tasks. To address reviewer 2XXj’s concerns, we provide both conceptual and experimental justifications.
>
> ---
>
> Conceptually, we want to emphasize that the main contribution of this work is to introduce Gaussian Memory Attention (GMA) as an alternative way (other than simple feature distillation) to integrate spatial and semantic features. This contribution is acknowledged by reviewer a6FD and reviewer RUFb. The references provided by reviewer 2XXj are robotics applications of representations obtained via naive distillation, which is orthogonal to our method. Specifically, we provide extensive experiments to show that this memory attention mechanism is more effective than naive distillation on various downstream visual discriminative tasks even when the same set of reference features are used to optimize the representation.
>
> Experimentally, due to the limited rebuttal timeframe and the methodological focus of our work to advocate an alternative paradigm to build representations, it is impractical to perform full-scale robotics experiments on all the provided references. Besides providing more results on 2D vision tasks, to motivate how our method can be applied to robotics tasks, we include a new set of real robot experiments to use GMA for the zero-shot grasping task setting proposed in LERF-TOGO [1], where the robot is tasked to grasp objects by parts specified via language to evaluate its part-level understanding ability.
>
> **[Q1]** *Real robot deployment with part-level understanding.*
>
> > *Geometry-based or semantic-part-based tasks using DINO or diffusion features, such as geometry-based object retrieval/grasping[1, 2], planning objects to the goal configuration[3].*
> >
>
> We enclose a new [[video: pickup screwdriver]](https://drive.google.com/file/d/1Oimra-s_nwakyoYtJbekJTb_mj4FNKJA/view?usp=sharing) on tabletop manipulation, the task is constituted of two steps:
>
> 1. Pick up the screwdriver on the handle.
> 2. Find the plate and put down the screwdriver on the plate.
>
> The first step requires the feature field to be able to be language-grounded and part-level, and the second step requires language grounding as well as the effective integration of M3 with low-level control algorithms.
>
> For the first and second steps, we also visualize the first-person view of the grounding masks ([[grounding: step 1]](https://drive.google.com/file/d/15O3rJ9k_S-5tTRld6zC-QctqZmfopjiM/view?usp=sharing), [[grounding: step 2]](https://drive.google.com/file/d/1PPG_OYxyDtQKmBKKF6i6o8oQGY8Z0zPQ/view?usp=sharing)) to locate the screwdriver and plate.
>
> **[Q2]** *Real robot deployment with language-guided task.*
>
> > *Language-guided tasks using CLIP feature, such as like "pick up the red cube and place it on the blue book." (example references: [4, 5])*
> >
>
> We show real robot deployment tasks with a language-guided query, below are the experiment results and language query.
>
> 1. [[video: pickup sponge]](https://drive.google.com/file/d/1v5JtroD3jYpvyWnoFNU5Vcm3Q3bcrA7r/view?usp=sharing) Pick up the sponge and place it into the sink.
> 2. [[video: pickup pineapple]](https://drive.google.com/file/d/1mWjn2_ssM1MvFZtIY0wzOjR28HiUNFBf/view?usp=sharing) Pick up the pineapple from the kitchen set and place it on to the table.
>
> In addition, we kindly share the perspective, that language-guided task performance is equivalent to grounding task as shown in the table below:
>
> | Dataset | Method | # Param | mIoU | cIoU | AP50 | AP60 |
> | --- | --- | --- | --- | --- | --- | --- |
> | Train | GT | - | 25.3 | 26.3 | 14.7 | 3.3 |
> | Train | F-3DGS | 61M | 24.2 | 24.3 | 16.3 | 7.1 |
> | Train | M3 | **35M** | **25.4** | **26.5** | **19.6** | **12.5** |
> | Playroom | GT | - | 25.6 | 24.2 | 9.6 | 3.0 |
> | Playroom | F-3DGS | 61M | 23.8 | 21.4 | 11.9 | 3.0 |
> | Playroom | M3 | **35M** | 23.1 | **23.1** | **11.9** | **5.9** |
> | Geisel | GT | - | 19.5 | 21.4 | 5.3 | 0.0 |
> | Geisel | F-3DGS | 61M | 19.0 | 20.4 | 14.1 | 1.2 |
> | Geisel | M3 | **35M** | **21.8** | **23.5** | **16.5** | **11.8** |
>
> References:
>
> 1. *Rashid, Adam, et al. "Language embedded radiance fields for zero-shot task-oriented grasping." CoRL. 2023.*

---

> > ### Author Response · Authors · 2024-11-28
> > **Response to Reviewer 2XXj (2/2)**
> >
> > **[Q3]** *Multi-step Long-Horizon task using multiple foundation models.*
> >
> > > *A multi-step task that combines different features, such as "navigate to the kitchen, locate the mug with a cat design, and bring it to the living room table." These examples would showcase how M3 can leverage different types of features for more complex robotic applications.*
> > >
> >
> > To evaluate the long-horizon task capability of M3, we build two memory scenes using M3.
> >
> > Scene 1: Shelf + Kitchen sets.
> >
> > Scene 2: Tabletop.
> >
> > For the multi-scene scenarios, we use different foundation models for each scene, specifically, we use CLIP and SigLIP embeddings. We use grasp sampling strategy as in LERF-TOGO [1] for grasping, and use keyframes used in training as navigation waypoints. Below are the demo videos for long-horizon tasks:
> >
> > 1. [[video: pickup sponge]](https://drive.google.com/file/d/1v5JtroD3jYpvyWnoFNU5Vcm3Q3bcrA7r/view?usp=sharing) Pick up the sponge and place it into the sink.
> > 2. [[video: pickup pineapple]](https://drive.google.com/file/d/1mWjn2_ssM1MvFZtIY0wzOjR28HiUNFBf/view?usp=sharing) Pick up the pineapple from the kitchen set and place it on to the table
> >
> > References:
> >
> > 1. *Rashid, Adam, et al. "Language embedded radiance fields for zero-shot task-oriented grasping." CoRL. 2023.*

---

> ### Author Response · Authors · 2024-12-01
>
> Dear Reviewer `2XXj`, it is approaching the rebuttal deadline on Dec 2nd at 11:59 AoE, it would be appreciated if you could take a look at our rebuttal. We want to re-emphasis on our contents:
>
> * Our main contribution is spatial memory via Gaussian splatting and foundation model embeddings.
> * We deploy our model on the real robot to prove the effectiveness of M3. The real-robot experiments provide additional verification of our approach, instead of the main contribution.
>
> Reviewer `a6FD` evaluates M3 with a focus on model architecture, while reviewer `RUFb` emphasizes feature representation learning. Additionally, reviewer `UxH3` concentrates on Gaussian representation. All three reviewers have provided positive feedback on our paper. We would greatly appreciate it if you could review our rebuttal materials and share any further comments. We are happy to address any remaining concerns.

---

> > ### Comment · Reviewer_2XXj · 2024-12-02
> > **Thank you!**
> >
> > I thank the detailed response from the authors, and have updated the rating.

---

> > > ### Author Response · Authors · 2024-12-02
> > >
> > > Thanks so much for reconsidering the scope of this paper, and the new opinion. We will try to make real robot experiments more solid for release.

---

### Official Review · Reviewer_UxH3 · 2024-11-03

**Soundness:** 2
**Presentation:** 3
**Contribution:** 3
**Rating:** 6
**Confidence:** 4

**Summary:**

The paper presents a method for integrating foundation model knowledge into a structural representation of the scene (here Gaussian Splatting). The approach is capable of integrating several foundation models' knowledge into Multimodal Memory. To this end, the authors propose Principal Scene Components, i.e. a method of feature reduction based on the assumption that multiple views of the scene contain redundant information. The paper evaluates the performance on a low level in feature space, and on a high level - through a downstream task, including robot deployment.

**Strengths:**

I believe that the authors of the paper correctly identified the drawbacks of 3D feature supervision through 2D feature maps (i.e. potential loss of information with dimensionality reduction). To this end, the paper proposes a compression scheme for extracted features named principal scene components (PSC). I believe this idea to be sound and interesting. I appreciate that the idea includes a flexible parameter of similarity threshold allowing the user to decide on memory/knowledge trade-off. In principle, the method seems to be trained in a similar way to prior works, making me believe that the proposed memory module can be used interchangeably to prior works, and could be applied to an arbitrary 3DGS work. Another benefit is a significant speed-up with respect to F-3DGS. Qualitative results seem convincing. I believe the use of robot manipulation as the downstream task is a good demonstration of the capabilities of the proposed approach.

**Weaknesses:**

While I believe the authors propose an interesting method, I see some weaknesses in the paper. Firstly, there are some drawbacks to the experimental setup:
- F-3DGS evaluated their approach through the task of novel view semantic segmentation. I see no reason why this could not be used in this paper as the experiment. This makes comparison between the two methods harder. Similarly, both F-3DGS and F-Splat do language-guided editing. Direct feature comparison is interesting and meaningful, however, we should also adhere to previously used evaluation protocols in order to facilitate fair comparisons.
- Similar to the previous point, I believe it would be beneficial to have some datasets between the proposed method and prior work in common.
- Further, the paper claims memory efficiency as a part of the method's benefits. Therefore an ablation and comparison to prior work from the memory perspective would be useful.
- There should be a study on whether there exists a correlation between performance, training time, and memory footprint (e.g. quantitative metrics for similar training time to F-3DGS - does M3 reach the same performance or does it plateau; also a comparison between models for the same feature size, etc.).

**Questions:**

- Do all the methods in comparison use the same underlying 3DGS model?
- Lines 48-49 mention both static scenes and time windows, could you explain that?
- Do you train reconstruction together, or do you start training with colour loss only? Alternatively, is the method sensitive to initialisation? With poor initialisation, the first iteration may be very noisy and rendered features very poor.
- Could you pinpoint the exact component that makes M3 training so much faster than F-3DGS?

---

> ### Author Response · Authors · 2024-11-28
> **Response to Reviewer UxH3 (1/2)**
>
> We sincerely appreciate reviewer `UxH3`for the positive comments on our contributions, especially for the proposed compression mechanisims in 2D-3D feature distillation. The suggestions in both the weakness and question sections are inspiring and encourage us to conduct more detailed analysis between our methods and previous works.
>
> ---
>
> **[Q1]** *Comparisons with previous methods (F-3DGS) on downstream tasks.*
>
> > *F-3DGS evaluated their approach through the task of novel view semantic segmentation. I see no reason why this could not be used in this paper as the experiment. This makes comparison between the two methods harder.*
> >
>
> Thanks for pointing this out, we agree that only showing the direct feature comparison is not convincing enough for performance comparison.  During the rebuttal, we included two new downstream tasks based on the reviewers’ feedback: grounding and retrieval, and we report their results here.
>
> In the table below, we show the downstream task performance of both F-3DGS and M3 in comparison with GT features in three datasets including Train, Playroom, and Geisel. The evaluation metrics cover both grounding and retrieval tasks, with metrics spanning IoU, AP, I2T, and T2I. We use MaskCLIP embeddings for grounding and SigLIP embeddings for retrieval.
>
> The table below shows the following insights:
>
> 1. With half the parameter size compared to F-3DGS, we not only achieve better results than F-3DGS by a reasonable margin (especially on AP50) but also outperform the original GT feature from MaskCLIP. This is caused by the Gaussian rendered feature usually having higher resolution as well as less noise.
> 2. For image retrieval, M3 performs significantly better than F-3DGS, while having relatively closer results with ground truth. Especially when we examine the top-1 results (I2T@1, T2I@1), M3 is much better than F-3DGS, this indicates that the feature map created via principal scene components does have a knowledge space more closely aligned with the original foundation model’s embedding space.
>
> | Dataset | Method | # Param | Grounding/CLIP |  |  |  | Retrieval/SigLIP |  |  |  |  |  |
> | --- | --- | --- | --- | --- | --- | --- | --- | --- | --- | --- | --- | --- |
> |  |  |  | mIoU | cIoU | AP50 | AP60 | I2T@1 | I2T@5 | I2T@10 | T2I@1 | T2I@5 | T2I@10 |
> | Train | GT | - | 25.3 | 26.3 | 14.7 | 3.3 | 81.5 | 97.3 | 100.0 | 71.0 | 89.4 | 92.1 |
> | Train | F-3DGS | 61M | 24.2 | 24.3 | 16.3 | 7.1 | 2.6 | 13.2 | 28.9 | 0.0 | 2.6 | 18.4 |
> | Train | M3 | **35M** | **25.4** | **26.5** | **19.6** | **12.5** | **55.2** | **84.2** | **92.1** | **52.6** | **84.2** | **92.1** |
> | Playroom | GT | - | 25.6 | 24.2 | 9.6 | 3.0 | 96.5 | 100.0 | 100.0 | 62.0 | 96.5 | 100.0 |
> | Playroom | F-3DGS | 61M | 23.8 | 21.4 | 11.9 | 3.0 | 79.3 | 96.6 | 96.6 | 31.0 | 79.3 | 89.7 |
> | Playroom | M3 | **35M** | 23.1 | **23.1** | **11.9** | **5.9** | 72.4 | **96.6** | **100.0** | **41.3** | 65.5 | 68.9 |
> | Geisel | GT | - | 19.5 | 21.4 | 5.3 | 0.0 | 100.0 | 100.0 | 100.0 | 60.0 | 85.7 | 91.4 |
> | Geisel | F-3DGS | 61M | 19.0 | 20.4 | 14.1 | 1.2 | 45.7 | 94.3 | 100.0 | 0.0 | 20.0 | 34.3 |
> | Geisel | M3 | **35M** | **21.8** | **23.5** | **16.5** | **11.8** | **100.0** | **100.0** | **100.0** | **71.4** | **85.7** | **94.2** |
>
> **[Q2]** *Comparison on the dataset with F-3DGS, and F-Splat.*
>
> > *Similar to the previous point, I believe it would be beneficial to have some datasets between the proposed method and prior work in common.*
> >
>
> Thank you for the constructive feedback. We agree that comparing M3 with previous works on common datasets is valuable. However, due to time constraints, we are still working on the implementation and will report the segmentation results on the dataset introduced in LERF shortly.
>
> **[Q3]** *Ablation study on memory efficiency in comparison with F-3DGS, F-Splat.*
>
> > *Further, the paper claims memory efficiency as a part of the method's benefits. Therefore an ablation and comparison to prior work from the memory perspective would be useful.*
> >
>
> The memory efficiency of the proposed method is achieved by using lower-dimensional feature embedding for each Gaussian primitive. For example, the CLIP feature is stored with only 16 dimensions, compared to 64 dimensions in F-Splat and 256 dimensions in F-3DGS (or 64 dimensions with speedup). Below is a detailed comparison of the parameter counts between our method and F-Splat/F-3DGS for the Garden scene with all six features:
>
> - Number of Gaussian Primitives: 138,766
> - Dimension of rendering parameters (e.g. position, SHS, etc.): 59
> - Feature dimension:
>     - Ours: 160
>     - F-Splat/F-3DGS w. speedup: 384
>     - F-3DGS w/o. speedup: 1536
>
> Parameter counts:
>
> - F-Splat/F-3DGS w. speedup: (59 + 384) * 138766 = 61 M
> - F-3DGS w/o. speedup: (59 + 1536) * 138766 = 221 M
> - Ours (Gaussian primitives): (59 + 160) * 138766  = 30 M
> - Ours (Principle scene component): 5 M
> - Ours (Total): 30 M + 5 M = 35 M

---

> > ### Author Response · Authors · 2024-11-28
> > **Response to Reviewer UxH3 (2/2)**
> >
> > **[Q4]** *Ablation study on memory footprint, performance, and training time.*
> >
> > > *There should be a study on whether there exists a correlation between performance, training time, and memory footprint (e.g. quantitative metrics for similar training time to F-3DGS - does M3 reach the same performance or does it plateau; also a comparison between models for the same feature size, etc.).*
> > >
> >
> > Yes, we believe that memory footprint, performance, and training time is a relatively important ablation for our work. As shown in Table. 3, we ablate the degree of freedom of principal queries in Gaussian primitives, this is the memory and performance ablation. Here, we add in the training time ablation in the format of iteration number. And the ablation on memory footprint, performance, and training time is complete. We evaluate our method on the Train dataset.
> >
> > | # Degree (PQ) | # Params↓ | # Iteration | CLIP |  |  |  | SigLIP |  |  |  | DINOv2 |  | SEEM |  | LLaMA3 |  | LLaMAv |  |
> > | --- | --- | --- | --- | --- | --- | --- | --- | --- | --- | --- | --- | --- | --- | --- | --- | --- | --- | --- |
> > |  |  |  | Cosine↓ | L2↓ | mIoU | AP50 | Cosine↓ | L2↓ | mIoU | AP50 | Cosine↓ | L2↓ | Cosine↓ | L2↓ | Cosine↓ | L2↓ | Cosine↓ | L2↓ |
> > | 8 x 6 = 48 | 14.8 M | 30k | 0.3256 | 0.2880 | 25.4 | 19.6 | 0.2913 | 0.5239 | 19.4 | 2.1 | 0.5755 | 1.7664 | 0.1672 | 0.2749 | 0.4504 | 0.0264 | 0.7229 | 53.6801 |
> > |  |  | 7k | 0.3290 | 0.2900 | 25.3 | 14.6 | 0.2938 | 0.5277 | 21.8 | 4.8 | 0.5845 | 1.7835 | 0.2058 | 0.3463 | 0.4517 | 0.0265 | 0.7243 | 53.7194 |
> > | 16 x 6 = 96 | 21.5 M | 30k | 0.3140 | 0.2800 | 25.7 | 19.0 | 0.2866 | 0.5172 | 24.3 | 10.3 | 0.5535 | 1.7239 | 0.1388 | 0.2247 | 0.4480 | 0.0261 | 0.7199 | 53.5893 |
> > |  |  | 7k | 0.3206 | 0.2842 | 25.3 | 20.6 | 0.2903 | 0.5227 | 23.2 | 8.1 | 0.5677 | 1.7513 | 0.1828 | 0.3056 | 0.4504 | 0.0263 | 0.7221 | 53.6545 |
> > | 32 x 6 = 192 | 34.8 M | 30k | 0.3043 | 0.2735 | 26.7 | 22.8 | 0.2814 | 0.5094 | 25.7 | 11.9 | 0.5318 | 1.6807 | 0.0972 | 0.1553 | 0.4401 | 0.0253 | 0.7159 | 53.4685 |
> > |  |  | 7k | 0.3132 | 0.2792 | 26.2 | 21.1 | 0.2866 | 0.5172 | 25.5 | 11.4 | 0.5515 | 1.7198 | 0.1269 | 0.2139 | 0.4436 | 0.0256 | 0.7195 | 53.5751 |
> > | 64 x 6 = 384 | 61.4 M | 30k | 0.2917 | 0.2650 | 28.4 | 23.9 | 0.2721 | 0.4957 | 28.5 | 13.5 | 0.5099 | 1.6358 | 0.0855 | 0.1321 | 0.4278 | 0.0241 | 0.7111 | 53.3236 |
> > |  |  | 7k | 0.3049 | 0.2734 | 28.1 | 23.9 | 0.2802 | 0.5079 | 27.8 | 13.5 | 0.5350 | 1.6870 | 0.1012 | 0.1676 | 0.4348 | 0.0248 | 0.7163 | 53.4816 |
> >
> > **[Q5]** *Are M3, F-3DGS, F-Splat in this paper using the same 3dgs model？*
> >
> > To ensure a fair comparison, all methods utilize the same underlying 3DGS models. For training data, we use identical image sets, training features, SFM point clouds, and camera parameters from the same scene dataset. Besides, all rendering hyperparameters (e.g. degrees of spherical harmonics) are kept consistent across methods.
> >
> > **[Q6]** *Confusion on L48-49 mentions both static scenes and time window.*
> >
> > Thanks for pointing out this potential confusion. Our method focuses on video clips captured along a camera trajectory in static scenes. The term “time window” in the original submission specifically refers to the sequence of frames within the video. To improve clarity, we have revised the introduction section of the paper.
> >
> > **[Q7]** *Do you train the color loss and feature loss together? Is the M3 sensitive to gaussian primitives initialization.*
> >
> > > *Do you train reconstruction together, or do you start training with colour loss only? Alternatively, is the method sensitive to initialisation? With poor initialisation, the first iteration may be very noisy and rendered features very poor.*
> > >
> >
> > We train the color and feature reconstruction simultaneously. During our previous experiments, we empirically observed that the method is not particularly sensitive to initialization.
> >
> > **[Q8]** *Why M3 trains so much faster than F-3DGS.*
> >
> > The faster training of M3 compared to F-3DGS is due to the smaller number of parameters and the loss design. First, M3 uses fewer parameters than F-3DGS (35M versus 61M, even with speedup applied to F-3DGS). Second, instead of computing pixel-wise distance loss on rendered features, our method adopts a point-based loss, sampling only 2000 points during each iteration.
> >
> > References
> >
> > 1. *Zhou, Shijie, et al. "Feature 3dgs: Supercharging 3d gaussian splatting to enable distilled feature fields." CVPR. 2024.*
> > 2. *Qiu, Ri-Zhao, et al. "Feature Splatting: Language-Driven Physics-Based Scene Synthesis and Editing." ECCV. 2024.*
> > 3. *Dong, Xiaoyi, et al. "Maskclip: Masked self-distillation advances contrastive language-image pretraining." CVPR. 2023.*
> > 4. *Zhai, Xiaohua, et al. "Sigmoid loss for language image pre-training." CVPR. 2023.*
> > 5. *Radford, Alec, et al. "Learning transferable visual models from natural language supervision." International conference on machine learning. PMLR, 2021.*

---

> ### Comment · Reviewer_UxH3 · 2024-11-30
> **Thanks for the rebuttal**
>
> I'd like to thank the authors for the rebuttal. I believe the extra experiment with a downstream task is very helpful. Personally, I'd suggest trying to fit it into the main paper. I also appreciate the additional ablation and the answers to my questions. Similarly, further experiments in the robot setup provided for another reviewer look convincing to me. Currently, the revised paper and a rebuttal text help to validate the claims of the paper. Given that, I increased my initial score.

---

> > ### Author Response · Authors · 2024-11-30
> >
> > We appreciate the prompt reply from the reviewer, `UxH3, ' and we have already enclosed the new experiments in the revision paper: https://openreview.net/pdf?id=XYdstv3ySl. We will continue to polish our paper in the rebuttal session. Thanks so much for your review again, those suggestions are very helpful!

---

### Official Review · Reviewer_RUFb · 2024-11-04

**Soundness:** 2
**Presentation:** 2
**Contribution:** 3
**Rating:** 6
**Confidence:** 4

**Summary:**

The paper introduces MultiModal Memory (M3), an integration of 3D Gaussian Splatting (GS) techniques with features from 2D Foundation Models (FMs) that replace standard feature distillation pipelines. The authors introduce two new concepts: a) principal scene components (PSC) which reduces the dimensionality of the FM features without sacrificing quality, thus reducing computational resources for training and storing features, and b) gaussian memory attention (GMA) which proposes to learn queries to memory as gaussian attributes (and not
the FM features themselves), thus dealing with the issue of misalignment between distilled and raw features of previous methods. The proposed memory can store features from multiple FMs, thus allowing a single GS structure to reconstruct multiple different features and solve many vision-language downstream tasks. The authors compare the quality of reconstructed features vs previous approaches in 4 scenes, do ablations in another 2 robotic-related scenes, provide comparisons with downstream tasks for 1 scene, and also provide a
demonstration with a mobile robot.

**Strengths:**

1. The introduction of Gaussian Memory Attention (GMA) is a novel and interesting way of linking Gaussian \ Splatting techniques with feature distillation. Previous works learn downsampled versions of the target features to reduce memory footprint, and decode them back with MLPs. The downsampled feature is the learned attribute in GS. With GMA, the learned attribute is a query to a memory, which can hold features from multiple foundation models. This allows a single trained GS to reconstruct features from many possible \ foundation models, instead of one GS per foundation model.

2. The introduction of Principal Scene Components (PSC) and the similarity reduction algorithm seems to be able to provide a good balance between memory footprint (#degrees stored in memory) vs. reconstruction quality, according to Table 1.

3. The authors integrate generative foundation models (Llama3v etc.) with their approach, while previous works only consider discriminative features (CLIP, DiNov2 etc.). The authors claim that the reconstructed features can be used directly as visual context to LLMs for generative tasks such as image captioning, which is I believe a great advancement of the distilled feature fields work, although they did not provide demonstrations for such applications.

**Weaknesses:**

1. The paper misses implementation details and qualitative results for downstream task applications. How to use the reconstructed FM features to do the tasks you show in your Fig.2, namely: image captioning, language grounding and retrieval? Please share some explanation of which feature is used for what task, and how, and provide qualitative comparisons with previous works F-3DGS and F-splat.

2. Lacking quantitative comparisons with baselines for downstream tasks. Only Table 4 contains comparisons for 1 scene with grounding task but lacks comparison with F-Splat baseline. I believe the authors need to perform more experiments for downstream tasks for more scenes and compare with both baselines, so the final performance-efficiency tradeoff from the introduction of PSC and attention memory can be appreciated.

 3. Also, currently the paper only shows reconstructed features from their M3 method. I believe the paper would benefit from a comparison of reconstructed features between previous approaches and M3, in order to appreciate qualitatively the effect of introducing the PSC method for reducing feature dimensionality and Gaussian Memory for decoding back the original feature.

 4. There is some confusion regarding the inclusion of the temporal aspect in the memory. The authors name their structure video memory and say that it holds medium-time video clips for static scenes. However, from the method and experiments, it seems that the temporal aspect has not been a focus of the paper, as the scenes are static, and the downstream tasks are only at the image or pixel level (captioning, grounding, etc.). Do the authors mean that the video contains the multi-view collection trajectory of an agent? If that is the case, I believe it should not claim to be a video memory and model the temporal aspect, as the temporal dimension in the video is just the dimension of new views of the same static scene.

 5.  I believe the paper would benefit from better clarity when presenting the method. The authors introduce many concepts by naming them (VG, KS, V) but do not provide input domains for such concepts therefore lacking precision. For example, in line 189, the authors write for a view V ∗ ∈ V, its V ∗ = {v 1 , v 2 , ..., v n }. What is the domain of V and V*? What are the intermediate segments vi? Later, in line 200, they write F ∗ (V) = {E 1 , E 2 , ...E n } for the foundation model features of a scene and say that n is the number of views. This seems a bit contradicting to me, as before they mentioned V* as a particular scene, and n the number of multi-granularity segments of that scene. Which one is it, n is a number of scenes or a number of segments inside a scene?

MINOR:
 - (line 250) as shown in?
 - Table 4 is mentioned before Tables 2 and 3. Maybe rename?
 - (line 062): [...] may not be be inherently 3D-consistent [...]
 - (line 064): [...] knowledge embbeded in [...]
 - (line 080): [...] outperforms what?

**Questions:**

1. How to use the reconstructed FM features to do the tasks you show in your Fig.2, namely: image captioning, language grounding, and retrieval? Please share some explanation of which feature is used for what task, and how, and provide qualitative comparisons with previous works F-3DGS and F-splat.

2. Can the authors provide qualitative examples comparing the reconstructed features of M3 with those from previous methods, such as F-3DGS and F-Splat? This would help assess the quality improvement gained through PSC and GMA.

3. Could the authors clarify if the term "video memory" implies the storage of multi-view data in static scenes or if there is an intention to extend this framework to dynamic scenes? If the latter, would the temporal dimension in the memory structure require additional model adjustments?

---

> ### Author Response · Authors · 2024-11-28
> **Response to Reviewer RUFb (1/2)**
>
> Rebuttal Resources to Reviewer `RUFb`: [[rebuttal pdf: RUFb]](https://drive.google.com/file/d/1wC3LCq8xlaliQkIpt4UgmjaYDlEmoHaZ/view?usp=sharing)
>
> ---
>
> We sincerely appreciate reviewer `RUFb` for the positive comments on our contributions, especially for the proposed GMA and PSC mechanisims. The suggestions in both the weakness and question sections are inspiring and encoure us to explore additional downstream tasks and conduct a more detailed analysis.
>
> ---
>
> **[Q1]** *Provide implementation details, qualitative, and quantitative evaluation for downstream tasks in comparison with F-3DGS, and F-Splat.*
>
> > *The paper misses implementation details and qualitative results for downstream task applications. […]
> Lacking quantitative comparisons with baselines for downstream tasks. […]
> […] Please share some explanation of which feature is used for what task, and how, and provide qualitative comparisons with previous works F-3DGS and F-splat.*
> >
>
> Both F-3DGS and F-Splat are distillation-based methods with similar architecture. Thus, because of time constraints, we only compare with F-3DGS. Empirically and according to the results in Table 1, F-3DGS usually has better performance than F-Splat.
>
> **Implementation Details:** We directly use the public codebase of F-3DGS. We use 64 dimensions in Gaussian primitives for F-3DGS, while we use 16 dimensions in Gaussian primitives for all foundation models except Llama3 and Llamav, which use 32 dimensions. For F-3DGS, they primarily use segmentation models and CLIP to build the raw feature, while in this work to have an apple-to-apple comparison, we only use the foundation model embeddings without semantic clustering. The input and testing data are the same for both M3 and F-3DGS.
>
> **Qualitative Results:** We compare the feature PCA visualization of M3 and F-3DGS on the Train and Geisel dataset with the foundation model CLIP and SigLIP. As clearly shown in the [[feature visualization]](https://drive.google.com/file/d/1i9zi6fykGp8q5xlcQgiIf_tsAobBy036/view?usp=sharing), the F-3DGS features appear smoother and more dispersed, with less defined edges and structures, resulting in a hazier and less spatially coherent representation where features overlap more. In contrast, the M3 features exhibit sharper boundaries, better-defined object structures, and localized color separations, indicating more compact, disentangled, and geometrically aligned feature representations.
>
> **Quantitative Results:** In the table below, we show the downstream task performance of both F-3DGS and M3 in comparison with GT in three datasets including Train, Playroom, and Geisel. The evaluation metrics cover both grounding and retrieval tasks, with metrics spanning IoU, AP, I2T, and T2I. We use MaskCLIP embeddings for grounding and SigLIP embeddings for retrieval. The table below shows the following insights:
>
> 1. With half parameter size in comparison with F-3DGS, we not only achieve better results than F-3DGS with a reasonable margin (especially on AP50) but also have better results in comparison to the original GT feature from MaskCLIP. This is caused by the Gaussian rendered feature usually having higher resolution as well as less noise.
> 2. For image retrieval, M3 performs significantly better than F-3DGS, while having relatively closer results with ground truth. Especially when we look at the top-1 results (I2T@1, T2I@1), M3 is much better than F-3DGS, this indicates that the feature map created via principal scene components does have closer knowledge space with the original foundation models embedding space.
>
> We report the result on grounding and retrieval here.
>
> | Dataset | Method | # Param | Grounding/CLIP |  |  |  | Retrieval/SigLIP |  |  |  |  |  |
> | --- | --- | --- | --- | --- | --- | --- | --- | --- | --- | --- | --- | --- |
> |  |  |  | mIoU | cIoU | AP50 | AP60 | I2T@1 | I2T@5 | I2T@10 | T2I@1 | T2I@5 | T2I@10 |
> | Train | GT | - | 25.3 | 26.3 | 14.7 | 3.3 | 81.5 | 97.3 | 100.0 | 71.0 | 89.4 | 92.1 |
> | Train | F-3DGS | 61M | 24.2 | 24.3 | 16.3 | 7.1 | 2.6 | 13.2 | 28.9 | 0.0 | 2.6 | 18.4 |
> | Train | M3 | **35M** | **25.4** | **26.5** | **19.6** | **12.5** | **55.2** | **84.2** | **92.1** | **52.6** | **84.2** | **92.1** |
> | Playroom | GT | - | 25.6 | 24.2 | 9.6 | 3.0 | 96.5 | 100.0 | 100.0 | 62.0 | 96.5 | 100.0 |
> | Playroom | F-3DGS | 61M | 23.8 | 21.4 | 11.9 | 3.0 | 79.3 | 96.6 | 96.6 | 31.0 | 79.3 | 89.7 |
> | Playroom | M3 | **35M** | 23.1 | **23.1** | **11.9** | **5.9** | 72.4 | **96.6** | **100.0** | **41.3** | 65.5 | 68.9 |
> | Geisel | GT | - | 19.5 | 21.4 | 5.3 | 0.0 | 100.0 | 100.0 | 100.0 | 60.0 | 85.7 | 91.4 |
> | Geisel | F-3DGS | 61M | 19.0 | 20.4 | 14.1 | 1.2 | 45.7 | 94.3 | 100.0 | 0.0 | 20.0 | 34.3 |
> | Geisel | M3 | **35M** | **21.8** | **23.5** | **16.5** | **11.8** | **100.0** | **100.0** | **100.0** | **71.4** | **85.7** | **94.2** |

---

> > ### Comment · Reviewer_RUFb · 2024-11-28
> > **Thanks for the clarifications.**
> >
> > I appreciate the authors for their clarifications and additional results that emphasize the contribution of their work. I am increasing my score.

---

> > > ### Author Response · Authors · 2024-11-29
> > > **Thanks for your reply!**
> > >
> > > We appreciated the prompt reply from the Reviewer `RUFb`, and we will continue to do more experiments in this rebuttal session : )

---

> ### Author Response · Authors · 2024-11-28
> **Response to Reviewer RUFb (2/2)**
>
> **[Q2]** *Feature reconstruction qualitative results in comparison with F-3DGS, F-Splat to show the effectiveness of PSC and GMA.*
>
> > *[…] I believe the paper would benefit from a comparison of reconstructed features between previous approaches and M3 […]
> Can the authors provide qualitative examples comparing the reconstructed features of M3 with those from previous methods, such as F-3DGS and F-Splat?*
> >
>
> In the previous question, we have shown the feature reconstruction qualitative results in comparison with F-3DGS ([[feature visualization]](https://drive.google.com/file/d/1i9zi6fykGp8q5xlcQgiIf_tsAobBy036/view?usp=sharing)). This visualization clearly shows that the M3 feature is more compact, disentangled, and geometrically aligned. In this question, we show more qualitative results on the construction trace from the principal query to the principal scene components and the final rendered feature.
>
> As shown in [[visual: gaussian memory attention trace]](https://drive.google.com/file/d/1iuWMaD67OAvNg9Yp4vF2JBVtVrMpK7d6/view?usp=sharing), we show the visualization of the principal query, psc, raw feature, and the final render feature. All the images, including the colored circles, are directly drawn by the algorithm. The trace clearly shows that the principal query will correctly attend to the principal scene components that correlate with the raw feature of the similar semantic/texture information for each spatial location. This shows that the claimed contribution in this work is evident.
>
> **[Q3]** *Terminology video memory is confusing for the static scene.*
>
> > *[…] Do the authors mean that the video contains the multi-view collection trajectory of an agent? If that is the case, I believe it should not claim to be a video memory and model the temporal aspect, as the temporal dimension in the video is just the dimension of new views of the same static scene.*
> >
>
> Thanks for pointing out this potential confusion. Our method focuses on video clips captured along a camera trajectory in static scenes. We agree with the reviewer that the video memory involves only the multi-view aspect without the temporal aspect. To address this, we have clarified the terminology in the introduction section of the revised paper.
>
> **[Q4]** *Lack of input domain for many newly introduced concepts.*
>
> > *I believe the paper would benefit from better clarity when presenting the method. The authors introduce many concepts by naming them (VG, KS, V) but do not provide input domains for such concepts therefore lacking precision. […]*
> >
>
> Thanks for pointing out this potential confusion. Below, we provide a detailed description of domain for the newly introduced concepts.
>
> Consider a view $\mathbf{V} \in \mathbb{R}^{h\times w,3}$ (where $h,w$ denote the pixel dimensions) in the 3D scene $\mathbf{V}$. A signle view is represented as $\mathbf{V} = ${$V^{1}, V^{2}, ..., V^{m}$}, where $V^{i} \in  \mathbb{R}^{p, 3}$ is the $i^{\text{th}}$ granularity of the view $V$, $p$ is the number of pixels, and $m$ denotes the total number of granularities.
>
> The foundation model features for each view are denoted as $\mathbf{F}(\mathbf{V}) = ${$\mathbf{E}^{1}, \mathbf{E}^{2}, ... \mathbf{E}^{n}$}  for each foundation model ( $\mathbf{F}$ ) and scene ($\mathbf{V}$), where $\mathbf{E}^{i} \in \mathbb{R}^{[h\times w,d]}$, and $d$ denotes the feature dimension. Note that we consistently use subscript to indicate each view (a total of $n$ views) and superscript to indicate each granularity or foundtaion model. (a total of $m$ foundatiaon models).
>
> In addition, we have added a section in the supplementary material, using illustration figures and text descriptions for the input domain and newly introduced concepts.
>
> **[Q5]** *Typos.*
>
> Thank you for pointing this out. We have already addressed all the minor comments for writing in the revised paper.
>
> References
>
> 1. *Zhou, Shijie, et al. "Feature 3dgs: Supercharging 3d gaussian splatting to enable distilled feature fields." CVPR. 2024.*
> 2. *Qiu, Ri-Zhao, et al. "Feature Splatting: Language-Driven Physics-Based Scene Synthesis and Editing." ECCV. 2024.*
> 3. *Dong, Xiaoyi, et al. "Maskclip: Masked self-distillation advances contrastive language-image pretraining." CVPR. 2023.*
> 4. *Zhai, Xiaohua, et al. "Sigmoid loss for language image pre-training." CVPR. 2023.*
> 5. *Radford, Alec, et al. "Learning transferable visual models from natural language supervision." International conference on machine learning. PMLR, 2021.*
> 6. *Dubey, Abhimanyu, et al. "The llama 3 herd of models." arXiv preprint arXiv:2407.21783 (2024).*

---

### Official Review · Reviewer_a6FD · 2024-11-05

**Soundness:** 3
**Presentation:** 3
**Contribution:** 3
**Rating:** 8
**Confidence:** 4

**Summary:**

The article proposes a novel approach for video memory combining structural information from Gaussian Splatting with semantic information gathered from foundation models. The main challenge in this approach is the high-dimensionality of the semantic features for encoding in Gaussian splats. This has been addressed in the literature by feature distillation, but the authors argue that this leads to inefficient misaligned distilled features.
The authors proposed a novel attention approach to extract so-called Principle Scene Component. Experiments on a number of datasets appear to show that the proposed approach compares well in terms of fidelity and efficiency with the state-of-the-art, supporting the authors' claims.

**Strengths:**

- The efficient integration of structural and semantic features is an important problem for AI, and the proposed method is an interesting contribution.
- The method appears to perform well on a broad range of measures and experiments.
- The robotic application is a nice addition.

**Weaknesses:**

- The Gaussian Memory Attention component is a central contribution of the paper but is rather briefly discussed in the paper. It would be good to provide a clearer rationale for why this method was chosen and why it achieves the claimed result? How does it compare to standard memory mechanisms (such as in transformers), and what could be the alternative implementations.
- Similarly, the article contains an extensive ablation study of the foundation models, but no analysis of the impact of this specific component on performance. It would be good to provide some evidence of that as an ablation study (eg, Gaussian Memory attention vs other attention implementation). This would provide valuable insight into the component's impact on overall performance.
- The robotic experiment is interesting but it is unclear how informative it is about real-world applicability. It would be good to discuss possible challenges in scaling the approach to more complex/realistic environments and tasks, and to clarify the limitations of the current implementation as well as the future work that would be required for a real-world application.

**Questions:**

See the questions above in the weaknesses section, in summary:
- Can you provide a rationale for Gaussian Memory Attention component and a comparison to standard approaches?
- Can you provide an ablation study to evidence the impact of this novel component on the system's performance?
- Can you provide a discussion of the limitations of the approach in the robotic scenario and the outstanding questions and limitations to be overcome for real-world application.

---

> ### Author Response · Authors · 2024-11-28
> **Response to Reviewer a6FD (1/2)**
>
> Rebuttal Resources to Reviewer **`a6FD`**: [[rebuttal pdf: a6FD]](https://drive.google.com/file/d/1xIQf89rQuM8DefgQruZMMulESz0gHo-g/view?usp=sharing), [[video: pickup screwdriver]](https://drive.google.com/file/d/1Oimra-s_nwakyoYtJbekJTb_mj4FNKJA/view?usp=sharing)
>
> ---
>
> We sincerely appreciate reviewer **`a6FD`** for the positive comments on our contributions, especially for the combination of spatial and semantic features. The suggestions in both the weakness and question sections are very inspiring and encourage us to explore further in the direction of the GMA architecture and beyond.
>
> ---
>
> **[Q1]** *Discuss the motivation and implementation details of Gaussian Memory Attention.*
>
> > *The Gaussian Memory Attention component is a central contribution of the paper but is rather briefly discussed in the paper. It would be good to provide a clearer rationale for why this method was chosen and why it achieves the claimed result?*
> >
>
> Thanks for pointing out the key components of M3 with the correct emphasis that we need more motivation for the Gaussian Memory Attention (GMA) module. The GMA module is designed with the following rational steps:
>
> - The principal query is a compressed index of the principal scene component ($PSC$).
> - The up-sampled principal query ($PQ$) should have the same dimensionality as the principal scene components ($PSC$) → [L298: $Q_p \times W_m$].
> - The up-sampled principal query ($PQ$) can directly compute the cosine similarity with principal scene components ($PSC$) → [L298:  $\text{softmax}\left(Q_p W_m \times PSC^T\right)$].
> - The weighted sum of the principal scene components ($PSC$) based on the similarity could create the final rendered feature ($R$) → [L298:  $\text{softmax}\left(Q_p W_m \times PSC^T\right) \times PSC$].
> - The final rendered feature is a direct sum of principal scene components so it should be in the knowledge space of the original foundation models’ embeddings.
>
> We clearly show in the visualization [[visual: gaussian memory attention trace]](https://drive.google.com/file/d/1iuWMaD67OAvNg9Yp4vF2JBVtVrMpK7d6/view?usp=sharing) that the Gaussian Memory Attention operates as described above.
>
> **[Q2]** *Compare Gaussian Memory Attention with classic transformer architecture.*
>
> > *How does it compare to standard memory mechanisms (such as in transformers)*
> >
>
> The scaled dot-product attention operator is computed as follows:
>
> $$
> \text{Attention}(Q, K, V) = \text{softmax}\left(\frac{QK^\top}{\sqrt{d_k}}\right)V
> $$
>
> While the Gaussian memory attention operator is computed as:
>
> $$
> \text{GMA}(Q_p, W_m, PSC) = \text{softmax}(Q_p W_m \times PSC^T) \times PSC
> $$
>
> While the attention mechanism is quite similar, the major difference is the transformer architecture has projection layers for $Q, K, V$, as well as the output (could be in a multi-head way).  This shifts the distribution of both the principal query and principal scene components leading to the output space out-of-distribution of the foundation model knowledge space. Thus, the implementation of GMA directly fits the 3D feature field reconstruction scenario.
>
> **[Q3]** *Potential Variants of Gaussian Memory Attention (GMA).*
>
> > *and what could be the alternative implementations.*
> >
> - The simplest variant of Gaussian Memory Attention is removing the memory bank (Principal Scene Components) and the weighted sum procedure, preserving only the projection. This variant is known as distillation. Our baselines F-3DGS, F-Splat are both distillation methods.
> - We currently implement a version of GMA with softmax temperature. In our experiments, softmax without temperature significantly hurt performance.
> - We appreciate the reviewer mentioning transformer architecture, which is quite similar to ours.
> - The PSC could be initialized with raw features with learnable parameters, while it is fixed currently.

---

> > ### Author Response · Authors · 2024-11-28
> > **Response to Reviewer a6FD (2/2)**
> >
> > **[Q4]** *Ablation study of the performance of GMA, transformer, and new potential variants*.
> >
> > > *Similarly, the article contains an extensive ablation study of the foundation models, but no analysis of the impact of this specific component on performance. It would be good to provide some evidence of that as an ablation study (eg, Gaussian Memory attention vs other attention implementation). This would provide valuable insight into the component's impact on overall performance.*
> > >
> >
> > Due to time constraints, we conduct ablation studies of GMA with Transformer architecture. We will implement new potential variants shortly.
> >
> > | Dataset | Method | Degree | CLIP |  | SigLIP |  | DINOv2 |  | SEEM |  | LLaMA3 |  | LLaMAv |  |
> > | --- | --- | --- | --- | --- | --- | --- | --- | --- | --- | --- | --- | --- | --- | --- |
> > |  |  |  | Cosine | L2 | Cosine | L2 | Cosine | L2 | Cosine | L2 | Cosine | L2 | Cosine | L2 |
> > | Train | GMA | 160 | 0.3141 | 0.2800 | 0.2815 | 0.5094 | 0.5323 | 1.6816 | 0.1391 | 0.2253 | 0.4400 | 0.0253 | 0.7159 | 53.46 |
> > |  | Transformer | 160 | 0.9871 | 0.5512 | 0.9271 | 1.1039 | 0.9277 | 2.1970 | 0.9711 | 1.1828 | 0.9849 | 0.0447 | 0.9912 | 58.39 |
> >
> > The table clearly shows that using Gaussian memory attention has much better performance than directly using transformer architecture. This is because the projection layers in the transformer would easily negatively impact the representation space of PSC, thus falling into a different knowledge space in comparison with foundation model embeddings.
> >
> > **[Q5]** *Real robot deployment on long-horizon tasks.*
> >
> > > *The robotic experiment is interesting but it is unclear how informative it is about real-world applicability. It would be good to discuss possible challenges in scaling the approach to more complex/realistic environments and tasks, and to clarify the limitations of the current implementation as well as the future work that would be required for a real-world application.*
> > >
> >
> > We enclose a new [[video: pickup screwdriver]](https://drive.google.com/file/d/1Oimra-s_nwakyoYtJbekJTb_mj4FNKJA/view?usp=sharing) on tabletop manipulation, the task is constituted of two steps:
> >
> > 1. Pick up the screwdriver on the handle.
> > 2. Find the plate and put down the screwdriver on the plate.
> >
> > The first step requires the feature field to be able to be language-grounded and part-level, and the second step requires language grounding as well as the effective integration of M3 with low-level control algorithms.
> >
> > For the first and second steps, we also visualize the first-person view of the grounding masks ([[grounding: step 1]](https://drive.google.com/file/d/15O3rJ9k_S-5tTRld6zC-QctqZmfopjiM/view?usp=sharing), [[grounding: step 2]](https://drive.google.com/file/d/1PPG_OYxyDtQKmBKKF6i6o8oQGY8Z0zPQ/view?usp=sharing)) to locate the screwdriver and plate.
> >
> > **[Q6]** *Potential limitations on real-robot deployment.*
> >
> > > *Can you provide a discussion of the limitations of the approach in the robotic scenario and the outstanding questions and limitations to be overcome for real-world application.*
> > >
> >
> > Although there are no direct burdens that link Gaussian feature representation with real-robot deployment, there are several difficulties we have faced when we do deploy the real robot experiments:
> >
> > - Although we have optimized the implementation of our spatial memory to reduce training time, the current implementation still requires 60 minutes to optimize all representations for a complete scene, as noted in Table 1 of the revised PDF. While spatial memory provides high-quality representations that enable robots to understand the static components of a scene, it is not suitable for real-time robotics tasks. A promising future direction is to semantically separate the static and dynamic parts of the scene to allow real-time local updates of the memory.
> > - Calibrating the Gaussian representation of absolute spatial location with the robot’s location is a non-trivial task.
> >
> > References
> >
> > 1. *Zhou, Shijie, et al. "Feature 3dgs: Supercharging 3d gaussian splatting to enable distilled feature fields." CVPR. 2024.*
> > 2. *Qiu, Ri-Zhao, et al. "Feature Splatting: Language-Driven Physics-Based Scene Synthesis and Editing." ECCV. 2024.*

---

> > > ### Comment · Reviewer_a6FD · 2024-12-02
> > >
> > > Thank you for the detailed response to my questions, that dispelled any concern I had about the article.

---

> > > > ### Author Response · Authors · 2024-12-02
> > > >
> > > > Thanks so much for your initial great comment, and we appreciate your review : )

---

### Author Response · Authors · 2024-11-28
**General Response [1/2]**

We thank the reviewers for the meaningful comments spanning model architecture (`a6FD`), feature representation (`RUFb`), gaussian representation (`UxH3`), and robotics (`2XXj`). ***These suggestions motivate us to extend our work to the next stage*.**

**→ Strength:** We appreciate the reviewers recognize M3’s performance improvements in both quality (`a6FD`, `RUFb`) and efficiency (`a6FD`, `UxH3`), and value its potential for real-world robotic applications (`a6FD`,`UxH3`,`2XXj`). ***Most importantly, all the reviewers believe that the proposed idea is interesting and novel:***

**[Novelty `a6FD`, `RUFb`, `UxH3`, `2XXj`]**: The efficient integration of structural and semantic features is an important problem, and the proposed method is an interesting contribution (`a6FD`). In addition, the proposed module Gaussian Memory Attention (GMA) is novel and interesting (`RUFb`), and correctly identified the drawbacks of 3D feature supervision through 2D feature maps (`UxH3`). The proposed Principal Scene Components (PSC) and the reduction algorithm provide a good memory footprint vs. reconstruction quality (`RUFb`), and this idea is believed to be interesting and sound (`UxH3`). Meanwhile, generative foundation models are included in M3, which I believe a great advancement of the distilled feature field work (**`RUFb`**).

**→ Weakness:** Although the novelty of M3 is well recognized, all the reviewers propose constructive experimental suggestions of the proposed idea. We sincerely apologize for not uploading the rebuttal earlier, but ***we have intensively addressed the reviewer’s questions and suggestions regarding quantitative/qualitative results, real robot experiments, motivations, and writing***.

**[Downstream Tasks Performance]** While the proposed M3 outperforms previous methods in feature reconstruction, we also aim to demonstrate its effectiveness on downstream tasks. Although we have shown some downstream task performance in the initial submission, during the rebuttal, we included a more comprehensive version of grounding and retrieval evaluation. Please refer to the supplementary material for the newly proposed dataset.

In the table below, we show the downstream task performance of both F-3DGS and M3 in comparison with GT features in three datasets including Train, Playroom, and Geisel. The evaluation metrics cover both grounding and retrieval tasks, with metrics spanning IoU, AP, I2T, and T2I. We use MaskCLIP embeddings for grounding and SigLIP embeddings for retrieval.

The table below shows the following insights:

1. With half the parameter size compared to F-3DGS, we not only achieve better results than F-3DGS by a reasonable margin (especially on AP50) but also outperform the original GT feature from MaskCLIP. This is caused by the Gaussian rendered feature usually having higher resolution as well as less noise.
2. For image retrieval, M3 performs significantly better than F-3DGS, while having relatively closer results with ground truth. Especially when we examine the top-1 results (I2T@1, T2I@1), M3 is much better than F-3DGS, this indicates that the feature map created via principal scene components does have a knowledge space more closely aligned with the original foundation model’s embedding space.

| Dataset | Method | # Param | Grounding/CLIP |  |  |  | Retrieval/SigLIP |  |  |  |  |  |
| --- | --- | --- | --- | --- | --- | --- | --- | --- | --- | --- | --- | --- |
|  |  |  | mIoU | cIoU | AP50 | AP60 | I2T@1 | I2T@5 | I2T@10 | T2I@1 | T2I@5 | T2I@10 |
| Train | GT | - | 25.3 | 26.3 | 14.7 | 3.3 | 81.5 | 97.3 | 100.0 | 71.0 | 89.4 | 92.1 |
| Train | F-3DGS | 61M | 24.2 | 24.3 | 16.3 | 7.1 | 2.6 | 13.2 | 28.9 | 0.0 | 2.6 | 18.4 |
| Train | M3 | **35M** | **25.4** | **26.5** | **19.6** | **12.5** | **55.2** | **84.2** | **92.1** | **52.6** | **84.2** | **92.1** |
| Playroom | GT | - | 25.6 | 24.2 | 9.6 | 3.0 | 96.5 | 100.0 | 100.0 | 62.0 | 96.5 | 100.0 |
| Playroom | F-3DGS | 61M | 23.8 | 21.4 | 11.9 | 3.0 | 79.3 | 96.6 | 96.6 | 31.0 | 79.3 | 89.7 |
| Playroom | M3 | **35M** | 23.1 | **23.1** | **11.9** | **5.9** | 72.4 | **96.6** | **100.0** | **41.3** | 65.5 | 68.9 |
| Geisel | GT | - | 19.5 | 21.4 | 5.3 | 0.0 | 100.0 | 100.0 | 100.0 | 60.0 | 85.7 | 91.4 |
| Geisel | F-3DGS | 61M | 19.0 | 20.4 | 14.1 | 1.2 | 45.7 | 94.3 | 100.0 | 0.0 | 20.0 | 34.3 |
| Geisel | M3 | **35M** | **21.8** | **23.5** | **16.5** | **11.8** | **100.0** | **100.0** | **100.0** | **71.4** | **85.7** | **94.2** |

---

> ### Author Response · Authors · 2024-11-28
> **General Response [2/2]**
>
> **[Long Horizon Real Robot Tasks]**: We introduce three new real robot deployment videos in the supplementary material, they contain (1) Multiple steps, and (2) Multiple scenes.
>
> 1. [[video: pickup sponge]](https://drive.google.com/file/d/1v5JtroD3jYpvyWnoFNU5Vcm3Q3bcrA7r/view?usp=sharing) Pick up the sponge and place it into the sink.
> 2. [[video: pickup pineapple]](https://drive.google.com/file/d/1mWjn2_ssM1MvFZtIY0wzOjR28HiUNFBf/view?usp=sharing) Pick up the pineapple from the kitchen set and place it on to the table.
> 3. [[video: pickup screwdriver]](https://drive.google.com/file/d/1Oimra-s_nwakyoYtJbekJTb_mj4FNKJA/view?usp=sharing) Pick up the pineapple from the kitchen set and place it on to the table.
>
> The third video requires our representation to be able to recognize part-level semantics, identifying the graspable part of the screwdriver. All the video is achieved by the language-guided queries, at the same time for those videos with two scenes, we use different foundation models on each scene.
>
> References
>
> 1. *Zhou, Shijie, et al. "Feature 3dgs: Supercharging 3d gaussian splatting to enable distilled feature fields." CVPR. 2024.*
> 2. *Dong, Xiaoyi, et al. "Maskclip: Masked self-distillation advances contrastive language-image pretraining." CVPR. 2023.*
> 3. *Zhai, Xiaohua, et al. "Sigmoid loss for language image pre-training." CVPR. 2023.*
>
> **[Paper Revision Updates]**: We have submitted a revised version of our paper, with changes highlighted in red. The updates include:
>
> - Added the downstream evaluation results of grounding and retrieval to address the feedback from Reviewer `RUFb`and Reviewer `2XXj`.
> - Added the ablation study on memory footprint, performance, and training time to address the feedback from Reviewer `UxH3`.
> - Added the clarification on terminology and notations and the visualization of GMA to address the feedback from Reviewer `a6FD`.
> - Added the new long-horizon real robot deployments results to address the feedback from Reviewer `a6FD` and Reviewer `2XXj`.
> - Added additional feature PCA visualization between different methods to address the feedback from Reviewer `RUFb`.
> - Revised the `M3 Preliminaries` section to enhance the clarity of notations based on feedback from reivewer `RUFb`.
> - Revised the abstract and introduction sections to address the feedback from Reviewer `RUFb`and Reviewer `UxH3`.
>
> We sincerely appreciate the reviewers' constructive feedback. We believe this revision addresses most comments, but we are willing to address any additional feedback.

---

### Meta-Review · Area_Chair_TUf8 · 2024-12-23

**Metareview:**

**Summary**

The paper presents 3D spatial multimodal memory (M3) which distills semantic information from a combination of 2D foundation models into 3D Gaussian splatting.  The different 2D foundations models provides features at different granularities.  Experiments compare the proposed model against two recent methods that distills 2D features into Gaussian splats (F-3DGS, F-Splat), showing that M3 is more efficient that F-3DGS and has better feature similarity than F-Splat.

The key contribution of the work is how to compress the information from a diverse set of 2D foundation models onto 3D Gaussian without losing important information.  To do so, the paper introduces 1) "principle scene components" (PSC) which provides a compressed memory bank of 2D features, and 2) Gaussian memory attention (which provides attention between the PSC and queries).



**Strengths**
- Integration of semantic information into 3D is important [a6FD]
- Proposed method is interesting and allows for integration of many foundation models [a6FD,RUFb,UxH3,2XXj]
- Experiments show the method is effective [a6FD]
- Qualitative results are convincing [UxH3], and interesting downstream task (robot manipulation) is presented [UxH3,a6FD]


**Weaknesses**
Reviewers initially noted the following weakneses:
- Poor writing and presentation with some aspects not clearly explained
  - More explanation of Gaussian Memory Attention needed [a6FD]
  - Many implementation and experimental details missing [RUFb]
  - Various typos [RUFb,2XXj]
- Weaknesses in experiments
  - Questions about informativeness of experiments [a6FD]
  - Missing comparisons with baselines [RUFb]
  - Missing analysis of memory, training time, and performance [UxH3]
- Limited discussion of challenges [a6FD]

Most of the above where addressed during the author response period.

In addition the AC note that the related work is overally broad and not that relevant, while missing discussion of the most relevant related work (especially slightly earlier methods than F-3DGS and F-Splat (see **Suggested Revisions** for details).

**Recommendation**
Overall reviewers are positive on this work.  The AC also believes that the proposed method of combining many different features into Gaussian Splats can be of interest to the community and thus recommend acceptance.

**Suggested Revisions**
- Please provide more discussion of related work that actually distills / projects features / information from 2D foundation models into 3D.  The related work section on Foundation Models and Scene Graph and Video Memory discuss work that is not that relevant (e.g. L097-105).  It is recommended that more space and discussion is dedicated to more relevant related work (see suggested references below).
- Please consistently bold best performing numbers in Table 1 and Table 2 (not just the entries where M3 is the best)
- Please use \citep for parenthetical references (e.g. "ConceptGraphs Gu et al. (2024)" => "ConceptGraphs (Gu et al. 2024)")
- Please check and fix caption for Figure 5: "Illustration of patch-level visual embedding extraction their applications"
- L360: "Table." => "Table"
- L374: "Baseline Implementation" => "Baseline Implementation."

**Suggested references**

With 3D Gaussian Splats:

[1] LangSplat: 3D Language Gaussian Splatting, Qin et al. CVPR 2024

[2] GS3LAM: Gaussian Semantic Splatting SLAM, Li et al. ACMMM 2024

[3] Semantic Gaussians: Open-Vocabulary Scene Understanding with 3D Gaussian Splatting, Guo et al. 2024

[4] CLIP-GS: CLIP-Informed Gaussian Splatting for Real-time and View-consistent 3D Semantic Understanding, Liao et al. 2024

With 3D Gaussian Splats (2 foundation models)

[5] FMGS: Foundation Model Embedded 3D Gaussian Splatting for Holistic 3D Scene Understanding, Zuo et al. IJCV 2024

With NeRF models:

[6] FeatureNeRF: Learning generalizable NeRFs by distilling foundation models [Ye et al. ICCV 2023]

[7] Decomposing NeRF for Editing via Feature Field Distillation [Kobayashi et al. NeurIPS 2022]

[8] Neural Feature Fusion Fields: 3D Distillation of Self-Supervised 2D Image Representations [Tschernezki et al. 3DV 2022]

With NeRF models (two foundation models)

[9] LERF: Language embedded radiance fields [Kerr et al. ICCV 2023],

[10] M2DNeRF: Multi-Modal Decomposition NeRF with 3D Feature Fields [Wang et al. 2024]

With 3D point clouds:

[11] PointCLIP: Point Cloud Understanding by CLIP [Zhang et al. CVPR 2022]

[12] Image2Point: 3D Point-Cloud Understanding with 2D Image Pretrained Models [Xu et al. ECCV 2022]

[13] OpenScene: 3D Scene Understanding with Open Vocabularies [Peng et al. CVPR 2023]

**Additional Comments On Reviewer Discussion:**

Initially the paper received mixed reviews with three reviewers tending negative with a score of 5 [RUFb,UxH3,2XXj] and one reviewer highly positive on the work [a6FD].

Most of the reviewers acknowledge the that the proposed multimodal memory is novel and an interest way to compress information from multiple foundation models in Gaussian Splats.  All reviewers also had similar concerns regarding missing details and lack of proper comparisons.  During the author response period, the authors revised the paper to include a more proper comparison (Table 2) and improved Table 4 to include information about the number parameters.  The paper was also revised to provide more details about the proposed method and some experimental settings.  The AC notes that the paper can be improved further (see suggested revisions), but overall reviewers were satisfied with the revisions, with the three negative reviewer raising their score to 6.

---

### Decision · Program_Chairs · 2025-01-22

Accept (Poster)